# Dual-Policy Architecture for Multi-Agent Exploration

## Abstract

When training an agent using Reinforcement Learning, an efficient exploration strategy is essential to achieve good results. Multi-Agent Reinforcement Learning introduces additional challenges that require efficient exploration in order to find a set of policies that is able to achieve the goal. This is caused by the fact that agents may depend on each other to be successful. State-of-the-art works combine the exploitation and exploration behaviour into a single policy. Instead, we propose the use of a dual-policy architecture, where we separate the exploration policy from the exploitation policy. We present two different ways to accomplish such a dual-policy architecture, Weighted-Q Dual-Policy (WQ-DP) and $\epsilon$-Sampled Dual-Policy ($\epsilon$S-DP). WQ-DP uses an approach more similar to previous works, using a weighted sum of the Q-values produced by the exploitation and the exploration policies to choose an action. $\epsilon$S-DP samples between the exploitation and exploration policy based on the $\epsilon$ parameter that varies during training. Our results show that agents that use a dual-policy architecture outperform agents that combine the exploitation and exploration policies. $\epsilon$S-DP shows the best results when comparing the tested architectures. Further experiments show that the policy sampling period in $\epsilon$S-DP greatly contributes to its superior performance.

## 1 Introduction

Exploration is a crucial aspect of Reinforcement Learning (RL). It is essential that the agent efficiently explores the environment to be able to find rewarding state-action pairs and converge to an effective policy. In theory, in order to converge to the optimal policy, each state-action pair needs to be visited an infinite number of times. However, in practice, it is often possible to learn an effective policy with more limited experience. In environments with a small and discrete state- and action-space a random exploration strategy will usually be sufficient to explore the environment and converge to a policy that achieves the goal. When the environment has a large or continuous state- and/or action-space, it may no longer be possible to visit all state-action pairs. This can make it harder to find rewarding state-action pairs, especially when rewards are sparse or when parts of the state-space are hard to reach. In those cases, a more efficient and strategic exploration approach can be necessary.

We provide a brief overview of environment properties that can make the exploration of an environment challenging in single agent RL:

- **Sparse Rewards** in the environment are one of the main reasons to require strategic exploration. If the reward is sparse, only a small subset of the state-space will result in a reward. If the agent does not observe any reward, it will not be able to learn a useful policy (Ladosz et al., 2022).

- **Noisy-TV** is the phenomenon where an agent gets stuck exploring a part of the state-space that mainly consists of noise. It is usually explained by the example of an agent exploring a house that contains a TV. On the TV, new random images are displayed every timestep, but no reward will be gained by watching the TV. In this scenario, the agent will keep seeing novel states by watching the TV but will make no progress in exploring other parts of the environment that could result in a reward (Burda et al., 2019).

- **Bottlenecks in the State-Space** are parts of the state-transition graph where there is only one or a few paths to another part of the state-space. This decreases the chances of reaching those areas making them harder to explore (Toquebiau et al., 2024).

- **Zero-incentive Dynamics** are bottlenecks in the state-space where the agent is not rewarded for getting past them. This means that important behaviours are not rewarded, making it harder for the agent to learn them (Molinghen et al., 2023).

In Multi-Agent Reinforcement Learning (MARL), these environment properties also make exploration more difficult. As the number of agents increases the number of joint action pairs will grow exponentially, making it increasingly complex to properly explore the environment. Additionally, because multiple agents influence the state-transition and reward, there are additional challenges that can make an environment difficult to explore. In literature, the following challenging environment properties have been presented:

- **Relative Overgeneralisation** describes the problem where a group of agents prefers a suboptimal Nash equilibrium instead of the optimal Nash equilibrium because the suboptimal Nash equilibrium is better when paired with arbitrary actions from the other agents, for example during exploration. It is only when all agents follow the optimal policy that the optimal Nash equilibrium will be the best option. (Wiegand & Jong, 2004; Wei & Luke, 2016).

- **Interdependence** is a property of the environment that indicates how much the agents depend on each other to reach the goal in the environment. In environments with low interdependence the agents can reach the goal by all individually completing their sub-goal. However, environments with high interdependence contain bottlenecks in the state-space. These bottlenecks require a high level of coordination between the agents to reach rewarding states (Molinghen et al., 2023).

- **Perfect Coordination** is required when the agents need to follow a specific sequence of actions to be successful and if deviation from this sequence by one single agent leads to early termination or a penalty (Molinghen et al., 2023).

These challenges have significant overlap. Interdependence can be used as an umbrella term which also describes both relative overgeneralisation and perfect coordination. A situation requiring perfect coordination can also be described as a situation with high interdependence. An environment that suffers from relative overgeneralisation also describes high interdependence and has a big overlap with perfect coordination.

The combination of these challenges with the challenges shared with single-agent RL create an even more complex environment for exploration. For example, the combination of zero-incentive dynamics and bottlenecks in the state-space that are caused by interdependence can create a situation where the agents are not rewarded for very important cooperative behaviours. This makes learning these behaviours extremely difficult.

In the state of the art (Pathak et al., 2017; Raileanu & Rocktäschel, 2020; Zhang et al., 2024; Toquebiau et al., 2024; Wang et al., 2019), the exploration behaviour will often be learned through a weighted intrinsic reward that is added to the environment reward and will encourage exploring the environment. This results in a single policy that tries to simultaneously maximize the environment reward and explore the environment. One of the main challenges in cooperative MARL is credit-assignment (Albrecht et al., 2024). Determining which action from which agent caused the reward is difficult because all agents receive the same team reward. Combining this team reward with an additional intrinsic reward further amplifies the credit-assignment problem. The agents will now have to determine their contribution to the combined reward instead of only the environment reward. Essentially, the goal here is to solve a multi-objective problem with two objectives, exploitation and exploration. Most of the state of the art reduces this to a single objective problem using a weighted sum of the two rewards. However, doing this presents some additional difficulties (Roijers et al., 2013). In stationary environments, the focus on exploration will be mostly at the beginning of training and decrease throughout training. This means that the weights of the weighted sum of rewards will also change throughout training. Therefore, the agents will have to find a policy in a non-stationary environment, which

is inherently more difficult. In addition, there will always be a delay between changing the weights and obtaining a policy that achieves the corresponding combined objective.

In this work, we investigate the use of dual-policy architectures, where we separate the exploitation and exploration policies. We present two dual-policy architecture implementations, Weighted-Q Dual-Policy (WQ-DP) and $\epsilon$-Sampled Dual-Policy ($\epsilon$S-DP). The two policies on their own will be easier to learn for the agents than if we combine them into a single policy since they can each focus on maximizing either the environment or the intrinsic exploration reward. The first approach, WQ-DP is an evolution of the approach that is the standard in the state of the art. However, instead of making a weighted sum of the exploitation and exploration reward, we use a weighted sum of the Q-values from the exploitation and exploration policies to determine the action. In the results, we see that separating the exploitation and exploration policy results in a clear performance improvement. The second approach improves upon this further by sampling which policy will be used to determine the action. This provides a lot more control over which policy is used and results in a significant performance improvement.

Next, we investigate different aspects of the exploration behaviour to see which aspects contribute to the success of the $\epsilon$S-DP approach. First, we investigate the policy sampling period that determines how often the agents can switch between the exploitation and exploration policy. Next, we analyse whether synchronising the choice between the exploitation and exploration policy across the different agents helps improve the performance of the agents. This means that the decision between exploration and exploitation will be the same for all agents. Finally, we look into an alternative intrinsic reward, based on Laplacian representations, to see whether the architecture remains effective when using different intrinsic rewards. The Laplacian representation contains information about the geometry of the state-transition graph, allowing us to calculate a distance between two states.[rev1] The goal of this intrinsic reward is to encourage the agents to move away from the initial state of the episode to explore areas further away.

In our experiments, we use environments that are designed to test the ability of our approaches to deal with the multi-agent exploration challenges that we explained as well as several of the single-agent challenges. The Laser Learning Environment (LLE) challenges the agents with sparse rewards, bottlenecks in the state-space, zero-incentive dynamics, high interdependence and requires perfect coordination while the relative overgeneralisation environment focuses on relative overgeneralisation and high interdependence.[rev1]

## 2 Related Work

Within the field of RL, there have been a variety of proposed exploration techniques. In this section, we give an overview of the state of the art of exploration. First, we look at techniques designed for single-agent RL, and then we investigate methods used specifically for MARL.

### 2.1 Single-Agent Exploration

There are many approaches to handle exploration within RL (Ladosz et al., 2022). In a lot of cases, some form of random exploration is employed such as $\epsilon$-greedy, variations such as temporally extended $\epsilon$-greedy (Dabney et al., 2020) or adding noise to the actions in Deep Deterministic Policy Gradients (DDPG) (Lillicrap et al., 2016). Similarly, entropy regularization is often used in policy gradients approaches to encourage stochastic actions (Willaims & Peng, 1991; Ahmed et al., 2019). However, these methods do not suffice in all environments. They waste a lot of environment interaction by repeating state-action sequences that are already known to be ineffective. In these more challenging environments, a strategic form of exploration is desired, for example by using an intrinsic reward.

Many approaches generate an intrinsic reward that aims to encourage exploratory behaviour. One way to define this intrinsic reward is to reward visiting unseen states. There are many ways to compose an intrinsic reward to promote novel states (Ladosz et al., 2022). Most methods are based on either prediction error (Pathak et al., 2017; Raileanu & Rocktäschel, 2020; Burda et al., 2019; Zhang et al., 2024), state counts (Tang et al., 2017; Martin et al., 2017; Ostrovski et al., 2017; Machado et al., 2020) or memory (Badia et al., 2020b; Fu et al., 2017).

One of the best known methods is Random Network Distillation (RND) (Burda et al., 2019). Here, the intrinsic reward is the Euclidean distance between the output of the predictor network and the target network. The target network is randomly initialised, and the predictor network is trained alongside the agent policy to match the output of the target network.

Zhang et al. (2024) adapt RND to create a new intrinsic reward (NovelD). They combine the RND value of the current and next timestep to obtain the novelty change resulting from a certain action. They also add an episodic state visit-count which aims to encourage varied experience within an episode. However, using state-counts limits this approach to environments with a discrete state space and can cause scalability problems.

Henaff et al. (2024) have a similar goal. Their method, Exploration via Elliptical Episodic Bonuses (E3B) encourages the agent to display diverse behaviour throughout an episode. However, their method does not rely on state counts but on clustering of state embeddings allowing the use of continuous state spaces and better scalability of the state space.

Many of these approaches focus on achieving novelty in the state. However, this can be dependent on how the state is represented. Some unimportant state changes may have a big influence on the state representation, as is seen in the Noisy-TV problem. In addition, they focus on how to compose an intrinsic reward, that improves the exploration performance of the agents. Less attention is given to how to combine this intrinsic reward with the environment reward.

The most common approach to incorporate exploration behaviour into an agent is to use a linear combination of the extrinsic "exploitation" reward and the intrinsic "exploration" reward (Pathak et al., 2017; Raileanu & Rocktäschel, 2020; Zhang et al., 2024). However, this results in a single policy that tries to both exploit and explore at the same time. In many situations we do not want this. Once training has finished, we often want a policy that focuses purely on exploitation or that does only a limited amount of exploration which is hard to control using only the intrinsic reward weight. Annealing the weight of the intrinsic reward throughout training also makes the environment non-stationary which makes the problem more complex. Another downside to this approach is that the weights of the rewards in the linear combination need to be tuned differently depending on the reward functions.

Burda et al. (2019) create two separate value heads and use episodic rewards for the exploitation head and non-episodic rewards for the exploration head. However, much of the policy will still be shared and therefore, the non-stationarity caused by the exploration reward will still impact the exploitation policy. Bagot et al. (2020) decouple exploitation and exploration behaviour. The agent consists of two policies, one for exploitation and one for exploration. The exploitation policy has an additional action to indicate that it wants to explore. By separating both policies, there is more control over the exploration and the agent retains the knowledge about exploration.

## 2.2 Multi-Agent Exploration

In the case of MARL, most methods use randomised exploration strategies such as $\epsilon$-greedy or entropy regularization. Hsu et al. (2024) investigate the use of other randomised exploration approaches in cooperative MARL. They present a framework to incorporate these randomised exploration techniques in MARL. They test their method using perturbed history exploration (Kveton et al., 2019) and Langevin Monte-Carlo exploration (Xu et al., 2022). Another way to achieve this is by combining the diverse experience of multiple approaches in a single shared replay buffer as is done by Majumdar et al. (2020). They show the benefits of enabling interaction between an evolutionary approach and a MARL approach.[rev1]

In MARL, there has also been a lot of research into learned exploration strategies. A first group of approaches does this without the use of an intrinsic reward. Multi-Agent Variational Exploration (MAVEN) (Mahajan et al., 2019) introduces a hierarchical policy to better explore the joint action space thereby counteracting the suboptimal exploration of QMIX.

Relational Representation for Multi-Agent Exploration (REMAX) (Ryu et al., 2022) is designed to generate useful initial states for the agent. This is done through the use of a variational graph autoencoder that encodes the state to a latent space. A surrogate model is then used to calculate an exploration score to

indicate how useful it is to explore this state. The method searches for latent representations that maximize the exploration score, translates them back to the corresponding states and uses these as initial states for the agent training. However, an important downside of this approach that we do not have control of the starting state in every environment. Liu et al. (2024) present a similar approach that identifies interesting states and uses an imagination model to compose a trajectory to reach those states, overcoming the downside of REMAX. The agents are initialised in these states using the imagined trajectory and $\epsilon$-greedy is used to explore further.

Liu et al. (2022) propose an exploration approach based on successor features (Barreto et al., 2017). They first train on a wide range of tasks. Each task produces new actor policies but uses the same critic that uses a global successor feature network. They can change the task that is learned by changing the successor feature weights within the critic. Therefore, they are able to explore a new task more effectively by selecting the best pretrained policies using the shared critic and evaluating them using the successor weights of the new task. However, this requires designing a curriculum of tasks that gradually increases in difficulty and similarity to the target task.

Most other methods in the state of the art use intrinsic rewards to encourage exploration. Wang et al. (2019) create an intrinsic reward that is based on the influence of actions on the reward function and transition function of other agents. Their work is similar to the work of Jaques et al. (2019), who create an intrinsic reward that is based on the influence on the policy of the other agents.

Toquebiau et al. (2024) use an intrinsic reward called Joint Intrinsic Motivation (JIM) composed of a combination of NovelD (Zhang et al., 2024) and E3B (Henaff et al., 2024). In addition to this intrinsic reward they encourage the agents to explore the joint observation space by using a global intrinsic reward calculated using the joint observation.

Zheng et al. (2024) also choose to use a combination of curiosity and episodic intrinsic rewards. For the curiosity part, they use an additional linear factorization module such as Value Decomposition Networks (VDN) (Sunehag et al., 2017) or QMIX (Rashid et al., 2018) to calculate the intrinsic reward, separate from the one used to determine the policy of the agents. The episodic intrinsic reward is designed to encourage the agents to replay highly rewarding sequences.

Similar to the state of the art in single agent exploration, most works on multi-agent exploration combine the exploration and exploitation policy into a single policy. They achieve this by training the policy using a total reward formed as a weighted sum of the exploration and exploitation reward. In their work, Böhmer et al. (2019) separate the exploration and exploitation policy. For their exploration policy they use a centralised policy that has been optimised to make it more scalable. Liu et al. (2021) present an approach to using a separated exploitation and exploration policy. At each timestep, they create a mixture of the exploration and exploitation policies to select an action from. The exploration policies are trained to reach under-explored states in a restricted state-space. During training, they gradually widen this restriction to explore more states. The work of both Böhmer et al. (2019) and Liu et al. (2021) are similar to the work presented in this paper. However, the focus of this paper is to present a thorough analysis of the performance of dual-policy architectures and provide insight into why they perform better than the weighted sum approach. Böhmer et al. (2019) will face scalability issues due to the centralised exploration policy. The work of Liu et al. (2021) is mostly focused on their approach to restrict the state space and does not explicitly show the contribution of the separated policies to the performance of the agents.

## 3 Background

In this section, we explain some background information about existing methods and concepts that we will be using in our experiments.

### 3.1 Decentralised Partially Observable Markov Decision Process

In this work, we consider cooperative multi-agent environments. These can generally be described as a Decentralised Partially Observable Markov Decision Process (dec-POMDP) or a Multi-Agent Markov Decision

Process (MMDP) if the environment is fully observable (Oliehoek & Amato, 2016). In a Decentralised Partially Observable Markov Decision Process (dec-POMDP), a set of agents ($\mathbb{A}$) acts in the environment. Each of the agents $a \in \mathbb{A}$ can perform an action. We denote the individual action of an agent $a$ as $u^a \in \mathbb{U}^a(s_t)^{\text{rev1}}$ and the joint action of all agents as $u \in \mathbb{U}(s_t)^{\text{rev1}}$. These actions influence the environment causing a transition from one state $s \in \mathbb{S}$ to another based on the transition function $P$. However, in a dec-POMDP, the agents do not observe the full state of the environment but only a limited observation $o \in \mathbb{O}$. To alleviate the partial observability, it can be useful in certain environments to use the observation history $\tau$ instead of only the current observation. We use $\tau$ to denote the joint observation history and $\tau_a$ to denote the individual observation history of agent $a$.$^{\text{rev1}}$ The agents receive a team reward $r$. In this work, we make a distinction between the extrinsic or environment reward $r_e$ and the intrinsic or exploration reward $r_i$.

## 3.2 Centralised Training Decentralised Execution

Centralised Training Decentralised Execution (CTDE) is a commonly used paradigm within MARL (Rashid et al., 2018; Foerster et al., 2018; Sunehag et al., 2017). The idea is that the training of the agents can be performed in a centralised way and only the execution needs to be decentralised. Using centralised training allows the use of more information and centralised components as long as it is not required for execution. For exploration, this paradigm can also be used. When using CTDE, we can use joint information for the calculation of the intrinsic reward as proposed by Toquebiau et al. (2024). We can also synchronise elements across agents such as when to explore and when to exploit, as we will explain in Section 4.1.2.

## 3.3 $\epsilon$-Greedy Dual-Policy

One of the most used exploration techniques is $\epsilon$-greedy. Here, we will choose a random action instead of following the learned policy with a chance of $\epsilon$. For consistency with the algorithm naming in the remainder of this paper we will refer to $\epsilon$-greedy as $\epsilon$-Greedy Dual-Policy ($\epsilon$G-DP). Using this approach, we essentially use a dual-policy architecture where the exploration policy is a random policy.

Algorithm 1 shows the overall process of using a dual-policy architecture. Here, interaction with the environment is described using the *Reset()* function which starts a new episode and provides an initial observation and set of possible actions and with the *Done()* function that indicates whether the episode has terminated. The *Step()* function takes a joint action and performs a transition in the environment providing a new observation history, reward and set of possible actions. We use a replay buffer where experience is stored using the *Store()* function and sampled with the *Sample()* function. We use $U(\mathbb{U}^a(s_t))$ to denote a uniform distribution over the set of possible actions $\mathbb{U}^a(s_t)$ for agent $a$ in the current state. $\mathbb{U}(s_t)$ describes the set of possible joint actions. *TrainAgentPolicies*() is a function to train the given policies on the given experience. The exact behaviour of this function depends on which MARL approach will be used.$^{\text{rev1}}$

## 3.4 Weighted-Rewards Single-Policy

An approach that is frequently used in literature to combine the extrinsic and intrinsic rewards into a single reward is using a weighted sum (Zhang et al., 2024; Toquebiau et al., 2024; Zheng et al., 2024).

$$r_{total} = r_e + \beta r_i \tag{1}$$

This method can be seen as a multi-objective reward function that aims to optimise both the exploitation objective and the exploration objective. Using a linear combination of the rewards of multiple objectives is a widely used method in this context (Rădulescu et al., 2019). The resulting agent will aim to simultaneously maximise the extrinsic and the intrinsic reward. Usually, the weight of the intrinsic reward will be annealed throughout training, to achieve a final agent that focuses on the extrinsic reward with no or limited attention to exploration. In the remainder of this paper we will refer to this approach as Weighted-Rewards Single-Policy (WR-SP).

Algorithm 2 shows the process of WR-SP. Here, interaction with the environment is described using the *Reset()* function which starts a new episode and provides an initial observation and set of possible actions

---

**Algorithm 1** $\epsilon$-Greedy Dual-Policy[rev1]

---

$\epsilon \in [0, 1]$
$Q \leftarrow$ joint Q-function

**for** # training iterations **do**
    **for** # episodes **do**
        $\tau_t, \mathbb{U}(s_t) \leftarrow Reset()$
        **while** not $Done()$ **do**
            $\overrightarrow{Q} \leftarrow Q(\tau_t)$
            $u_t \leftarrow EpsilonGreedy(\overrightarrow{Q}, \mathbb{U}(s_t))$
            $\tau_{t+1}, r_t, \mathbb{U}(s_{t+1}) \leftarrow Step(u_t)$
            $\text{Store}(\tau_t, u_t, \tau_{t+1}, r_t)$
            $\tau_t \leftarrow \tau_{t+1}$
            $\mathbb{U}(s_t) \leftarrow \mathbb{U}(s_{t+1})$
        **end while**
    **end for**
    $\tau_t, u_t, \tau_{t+1}, r_t \leftarrow Sample()$
    $Q \leftarrow TrainAgentPolicies(Q, \tau_t, u_t, r_t, \tau_{t+1})$
**end for**

**function** EPSILONGREEDY($\overrightarrow{Q}, \mathbb{U}(s_t)$)
    **for all** $a \in \mathbb{A}$ **do**
        $u_t^a \leftarrow \begin{cases} \underset{u'^a}{\text{argmax}}(\overrightarrow{Q}) & \text{with probability of } (1 - \epsilon) \\ \sim U(\mathbb{U}^a(_t)) & \text{with probability of } \epsilon \end{cases}$
    **end for**
    $u_t \leftarrow [u_t^1, \ldots, u_t^N]$
    **return** $u_t$
**end function**

---

and with the *Done()* function that indicates whether the episode has terminated. The *Step()* function takes a joint action and performs a transition in the environment providing a new observation history, reward and set of possible actions. We use a replay buffer where experience is stored using the *Store()* function and sampled with the *Sample()* function. We use $U(\mathbb{U}^a(s_t))$ to denote a uniform distribution over the set of possible actions $\mathbb{U}^a(s_t)$ for agent $a$ in the current state. $\mathbb{U}(s_t)$ describes the set of possible joint actions. *ExplorationRewardCalculator()* is a function to calculate the intrinsic reward from the given experience and *TrainAgentPolicies()* is a function to train the given policies on the given experience. The exact behaviour of these functions depends on which intrinsic reward and which MARL approach will be used.[rev1]

---

**Algorithm 2** Weighted-Rewards Single-Policy[rev1]

---

$\epsilon \in [0, 1]$
$\beta \in \mathbb{R}_0^+$
$Q \leftarrow$ joint Q-function

**for** # training iterations **do**
    **for** # episodes **do**
        $\tau_t, \mathbb{U}(s_t) \leftarrow Reset()$
        **while** not *Done()* **do**
            $\overrightarrow{Q} \leftarrow Q(\tau_t)$
            $u_t \leftarrow EpsilonGreedy(\overrightarrow{Q}, \mathbb{U}(s_t))$
            $\tau_{t+1}, r_{e,t}, \mathbb{U}(s_{t+1}) \leftarrow Step(u_t)$
            $Store(\tau_t, u_t, \tau_{t+1}, r_{e,t})$
            $\tau_t \leftarrow \tau_{t+1}$
            $\mathbb{U}(s_t) \leftarrow \mathbb{U}(s_{t+1})$
        **end while**
    **end for**
    $\tau_t, u_t, \tau_{t+1}, r_{e,t} \leftarrow Sample()$
    $r_{total,t} \leftarrow WeightedRewards(r_{e,t}, \tau_t, u_t, \tau_{t+1})$
    $Q \leftarrow TrainAgentPolicies(Q, \tau_t, u_t, r_{total,t}, \tau_{t+1})$
**end for**

**function** WEIGHTEDREWARDS($r_{e,t}, \tau_t, u_t, \tau_{t+1}$)
    $r_{i,t} \leftarrow ExplorationRewardCalculator(\tau_t, u_t, \tau_{t+1})$
    $r_{total,t} \leftarrow r_{e,t} + \beta r_{i,t}$
    **return** $r_{total,t}$
**end function**

---

### 3.5 Joint Random Network Distillation

In this work, we use a modified version of episodic RND as intrinsic reward. RND (Burda et al., 2019) is a well established method for single agent exploration (Ladosz et al., 2022; Badia et al., 2020a;b). To calculate the intrinsic reward, it uses two models. One model has random fixed parameters while the other gets gradually updated to match the output of the first model. The intrinsic reward is calculated as the mean square error between the output of both models. If we were to calculate the reward in a multi-agent case, we would obtain the following individual reward for each agent:

$$r_{i,t}^a = ||\phi(o_{t+1}^a) - \phi'(o_{t+1}^a)||^2 \tag{2}$$

The input for these models is the observation of a single agent. However, to achieve joint exploration, we would like to calculate a joint reward for all the agents. Therefore, as proposed by Toquebiau et al. (2024), we will be using the joint observation as input. This allows us to explore the joint observation space instead of the individual observation space. It also provides us with a joint exploration reward (Joint Random Network

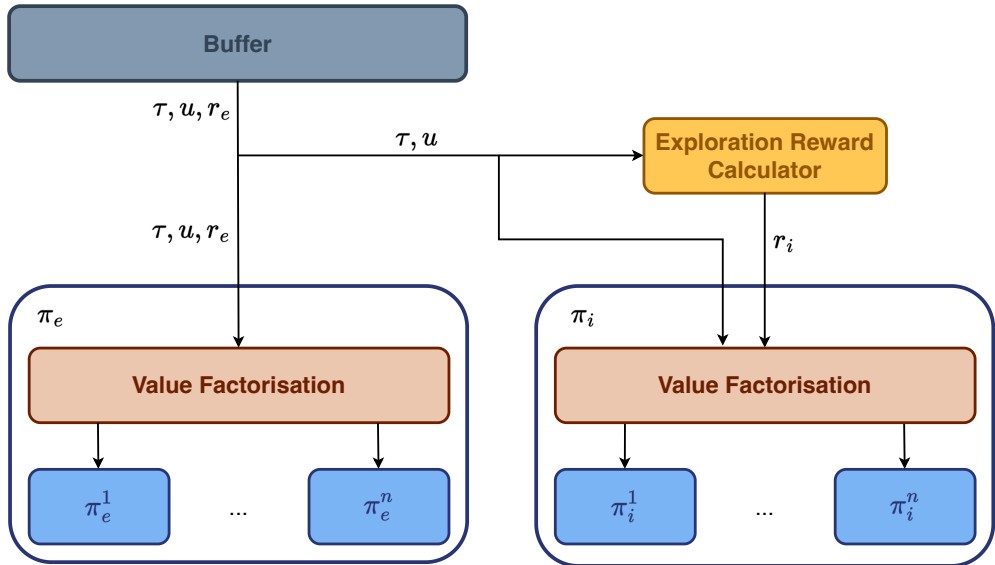

Figure 1: Dual-Policy Architecture at training time. The exploitation policies and exploration policies are trained using separate rewards on the same experience sampled from a replay buffer. The reward for the exploration policies is calculated using an Exploration Reward Calculator.[rev1]

Distillation (JRND)) instead of an individual exploration reward. We compose the joint observation of the agents by concatenating their individual observations.[rev1]

$$r_{i,t} = ||\phi(o_{t+1}) - \phi'(o_{t+1})||^2 \tag{3}$$

## 4 Methods

In this section, we present our methods. We will first look into the dual-policy approaches that we will investigate. Afterwards, we describe an alternative option for the intrinsic reward.

### 4.1 Dual-Policy Architecture

In order to isolate the exploration and exploitation behaviour, we split the policy of each agent into a separate exploitation and exploration policy. The exploitation policies ($\pi_e$) are trained using the extrinsic reward while the exploration policies ($\pi_i$) are trained using the intrinsic reward. This separation ensures that the exploitation policy is not contaminated with the exploration behaviour. Secondly this also allows us to differ learning hyperparameters.

Figure 1 shows the architecture of the agents during training. We group all exploitation policies together and train them using the extrinsic reward. For training, we can choose which multi-agent learning method to use. In this work, we focus on Q-learning approaches combined with a value factorisation approach such as VDN (Sunehag et al., 2017) or QMIX (Rashid et al., 2018). Q-learning methods are off-policy and therefore do not require any additional modifications when we train the policies on experience gathered using a different policy. Our proposed split architecture can also work with policy gradient or actor critic approaches. In this case, an off-policy variant of the policy gradient update rule needs to be used (Degris et al., 2012). This has previously been presented by Vanneste et al. (2023) using a uniform exploration policy and Liu et al. (2021) using a learned exploration policy.

We train the exploration policies using the intrinsic reward and using the same approach as described for the exploitation policies. The exploitation and exploration policies are trained on the same experience sampled

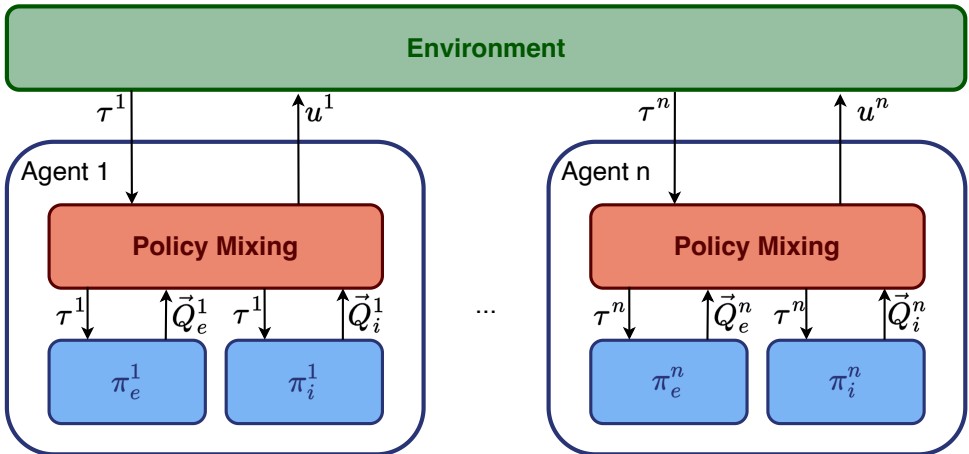

Figure 2: Dual-Policy Architecture when gathering experience. Each agent has two policies that produce a vector of Q-values given an observation history. Using a policy mixing approach a final action is determined based on these Q-values.[rev1]

from a shared replay buffer. The only difference is that an exploration reward calculator will calculate a different team reward ($r_i$) to train the exploration policies.

In Figure 2, the architecture that is used during experience gathering can be seen. Here, we group the policies trained using the architecture in Figure 1 by the agent it corresponds with. Each agent has two separate policies, one for exploitation and one for exploration. We calculate Q-values for both policies to determine the final action. Selecting the final action can be done in a variety of ways. In this work, we look at two variations.

In addition to the exploration policy, we still require that some actions are randomised for effective exploration since both policies are fully deterministic when using a Q-learning approach. Therefore, we still use $\epsilon$-greedy as well to make sure the agents always encounter some experience outside the focus of either policy.

Algorithm 3 shows the overall process of using a dual-policy architecture. Here, interaction with the environment is described using the *Reset()* function which starts a new episode and provides an initial observation and set of possible actions and with the *Done()* function that indicates whether the episode has terminated. The *Step()* function takes a joint action and performs a transition in the environment providing a new observation history, reward and set of possible actions. We use a replay buffer where experience is stored using the *Store()* function and sampled with the *Sample()* function. We use $U(\mathbb{U}^a(s_t))$ to denote a uniform distribution over the set of possible actions $\mathbb{U}^a(s_t)$ for agent $a$ in the current state. $\mathbb{U}(s_t)$ describes the set of possible joint actions. *ExplorationRewardCalculator()* is a function to calculate the intrinsic reward from the given experience and *TrainAgentPolicies*() is a function to train the given policies on the given experience. The exact behaviour of these functions depends on which intrinsic reward and which MARL approach will be used. In addition, the *PolicyMixing()* function calcultates the final action that will be performed based on the Q-values provided by both policies. In the following sections we provide two options for this function.[rev1]

### 4.1.1 Weighted-Q Dual-Policy

The first approach to combine both policies, Weighted-Q Dual-Policy (WQ-DP),[rev1] is the most similar to WR-SP. Here, before we perform an argmax operation to determine the action, we calculate a linear combination of the Q-values of both policies.

$$Q^a_{total}(\tau^a_t, u^a_t) = Q_e{}^a(\tau^a_t, u^a_t) + \beta Q_i{}^a(\tau^a_t, u^a_t) \tag{4}$$

$$\pi^a(\tau^a_t) = argmax_{u^a_t}(Q^a_{total}(\tau^a_t, u^a_t)) \tag{5}$$

---

**Algorithm 3** Dual-Policy MARL[rev1]

---

$Q_e \leftarrow$ joint exploitation Q-function
$Q_i \leftarrow$ joint exploration Q-function

**for** # training iterations **do**
    **for** # episodes **do**
        $\tau_t, \mathbb{U}(s_t) \leftarrow Reset()$
        **while** not $Done()$ **do**
            $u_t \leftarrow DualPolicyExperienceGathering(Q_e, Q_i, \tau_t, \mathbb{U}(s_t))$
            $r_{e,t}, \tau_{t+1}, \mathbb{U}(s_{t+1}) \leftarrow Step(u_t)$
            $Store(\tau_t, u_t, \tau_{t+1}, r_{e,t})$
            $\tau_t \leftarrow \tau_{t+1}$
            $\mathbb{U}(s_t) \leftarrow \mathbb{U}(s_{t+1})$
        **end while**
    **end for**
    $\tau_t, u_t, \tau_{t+1}, r_{e,t} \leftarrow Sample()$
    $Q_e, Q_i \leftarrow DualPolicyTraining(Q_e, Q_i, \tau_t, u_t, \tau_{t+1}, r_{e,t})$
**end for**

**function** DUALPOLICYTRAINING$(Q_e, Q_i, \tau_t, u_t, r_{e,t}, \tau_{t+1})$
    $r_{i,t} \leftarrow ExplorationRewardCalculator(\tau_t, u_t, \tau_{t+1})$
    $Q_e \leftarrow TrainAgentPolicies(Q_e, \tau_t, u_t, r_{e,t}, \tau_{t+1})$
    $Q_i \leftarrow TrainAgentPolicies(Q_i, \tau_t, u_t, r_{i,t}, \tau_{t+1})$
    **return** $Q_e, Q_i$
**end function**

**function** DUALPOLICYEXPERIENCEGATHERING$(\tau_t, \mathbb{U}(s_t))$
    $\overrightarrow{Q}_e \leftarrow Q_e{}^a(\tau_t)$
    $\overrightarrow{Q}_i \leftarrow Q_i(\tau_t^a)$
    $u_t \leftarrow PolicyMixing(\overrightarrow{Q}_e, \overrightarrow{Q}_i, \mathbb{U}(s_t))$
    **return** $u_t$
**end function**

---

This approach is very similar to WR-SP but has the advantage that the exploitation and exploration policies are decoupled and do not influence each other. However, it does not solve all challenges. The weight $\beta$ that we use to combine the Q-values is a hyperparameter that is not straightforward to tune and is specific to each reward function combination. This makes it hard to use. Using a linear combination causes the agents to perform actions that maximise the combined Q-value, but the agents will never act fully according to either policy. This is not ideal as we may not explore the areas that are prioritised by the exploration and may not perform the optimal actions determined using the exploitation policy either.

Algorithm 4 gives an overview of the process of choosing the final action using WQ-DP. Here, $U(\mathbb{U}^a(s_t))$ denotes a uniform distribution over the set of possible actions $\mathbb{U}^a(s_t)$ for agent $a$ in the current state.[rev1]

---

**Algorithm 4** Policy Mixing: Weighted-Q Dual-Policy[rev1]

$\epsilon \in [0, 1]$
**function** POLICYMIXING($\overrightarrow{Q}_e, \overrightarrow{Q}_i, \mathbb{U}(s_t)$)
    **for** $a \in \mathbb{A}$ **do**
$$u_t^a \leftarrow \begin{cases} \underset{u'^a}{\mathrm{argmax}}(\overrightarrow{Q}_e^a + \beta\overrightarrow{Q}_i^a) & \text{with probability of } (1 - \epsilon) \\ \sim U(\mathbb{U}^a(s_t)) & \text{with probability of } \epsilon \end{cases}$$
    **end for**
    $u_t \leftarrow [u_t^1, \ldots, u_t^N]$
    **return** $u_t$
**end function**

---

### 4.1.2 $\epsilon$-Sampled Dual-Policy

The second variation, $\epsilon$-Sampled Dual-Policy ($\epsilon$S-DP) tackles these concerns. Here, we make a discrete decision between the actions chosen by each of the policies. We decide between these two policies similarly to $\epsilon$-greedy. We use an $\epsilon$-value between 0 and 1 to indicate the probability of using the exploration policy. During training, we can change the $\epsilon$-value to allow the agents to explore more or less. This approach has a lot of similarities with the work presented by Böhmer et al. (2019), where they also sample between an exploitation or exploration policy. They, however, use a centralised exploration policy.

Combining both policies using this approach has several advantages. First, we can control the exploration using a more intuitive and generalisable hyperparameter. The $\epsilon$-value directly controls how much the agents will explore the environment. The weights we use for WQ-DP or WR-SP are much harder to tune and will not generalise to other reward functions.

Another advantage is that we can control how often we switch between the two policies. In many environments, it makes sense to allow the exploration and exploitation policies to act for longer than one timestep. This can allow the agents to explore or exploit further and reach better states than when switching between policies every timestep.

Algorithm 5 describes the process of choosing an action using $\epsilon$S-DP. Here, $U(\mathbb{U}^a(s_t))$ denotes a uniform distribution over the set of possible actions $\mathbb{U}^a(s_t)$ for agent $a$ in the current state. We use $\epsilon_{macro}$ to describe the probability of choosing exploration instead of exploitation and $\epsilon_{micro}$ to describe the probability of choosing a random action instead of using the exploration policy when in exploration mode. In this algorithm, we show $\epsilon$S-DP in the mixed configuration (See section 6.3)). The process for the other configuration options are included in Appendix A.1.[rev1]

### 4.2 Laplacian Representation Reward

The combination of zero-incentive dynamics and interdependence between agents can create bottlenecks in the state-space that are very difficult for the agents to learn to get past. We want the exploration policy to be incentivised to find a way past these bottlenecks. Intuitively, it makes sense to reward the agent for reaching areas far away from the initial state. By providing a dense reward that increases the further away from the initial state the agent gets, the agent will be encouraged to get past the bottlenecks in the state-space.

**Algorithm 5** Policy Mixing: $\epsilon$-Sampled Dual-Policy (mixed configuration (See section 6.3))[rev1]

$\epsilon_{macro} \in [0,1]$

$\epsilon_{micro} \in [0,1]$

$l \leftarrow$ policy sampling period (see Section 6.2)

$i \leftarrow l$

$choice^a \leftarrow$ "" for $a \in \mathbb{A}$

**function** POLICYMIXING($\overrightarrow{Q}_e$, $\overrightarrow{Q}_i$, $\mathbb{U}(s_t)$)

    $i \leftarrow i + 1$

    $choice\_micro \leftarrow \begin{cases} "policy" & \text{with probability of } (1 - \epsilon_{micro}) \\ "random" & \text{with probability of } \epsilon_{micro} \end{cases}$

    **for** $a \in \mathbb{A}$ **do**

        **if** $i > l$ **then**

            $choice^a \leftarrow \begin{cases} "exploitation" & \text{with probability of } (1 - \epsilon_{macro}) \\ "exploration" & \text{with probability of } \epsilon_{macro} \end{cases}$

        **end if**

        $u_t^a \leftarrow \begin{cases} \underset{u'^a}{\operatorname{argmax}}(\overrightarrow{Q}_e^a) & \text{if } choice^a == "exploitation" \\ \underset{u'^a}{\operatorname{argmax}}(\overrightarrow{Q}_i^a) & \text{if } choice^a == "exploration" \text{ and if } choice\_micro == "policy" \\ \sim U(\mathbb{U}^a(s_t)) & \text{if } choice^a == "exploration" \text{ and if } choice\_micro == "random" \end{cases}$

    **end for**

    **if** $i > l$ **then**

        $i \leftarrow 0$

    **end if**

    $u_t \leftarrow [u_t^1, \ldots, u_t^N]$

    **return** $u_t$

**end function**

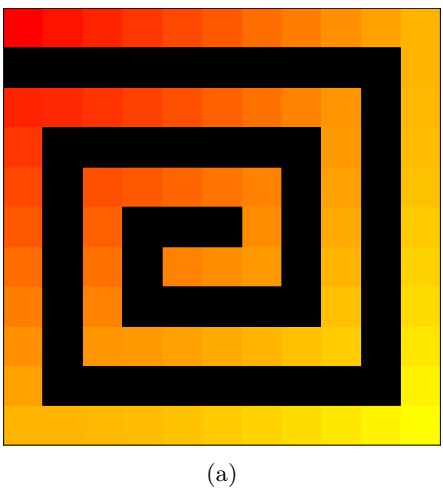
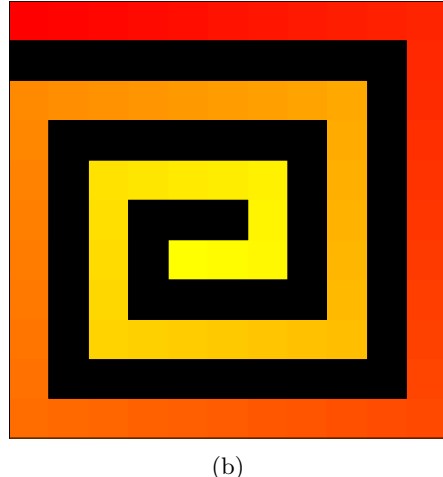

(a)                                                              (b)

Figure 3: Comparison between (a) L2 distance and (b) distance based on the number of steps it takes to reach a location. This shows that a metric taking into account the dynamics of the environment is more useful as intrinsic reward, encouraging the agents to move further away from the initial state.[rev1]

However, it is important to note that the use of such a reward will not be effective in all environments. It targets environments where the state-space contains bottlenecks. It will, however, not be as effective in environments such as the Multi-Particle Environment (MPE) where the agents can move infinitely away from their starting position without gaining useful experience.[rev2]

Determining the distance between two states in a Markov Decision Process (MDP) is not straightforward in most environments. For example, using the L2 distance in a 2D environment will never take walls or obstacles into account and will therefore not be sufficient for an intrinsic reward especially if we want to overcome bottlenecks in the state space. Determining a metric that is able to take these aspects into account requires an understanding of the transition function of the MDP. Figure 3 shows the difference between L2 distance and a distance metric based on the number of steps it takes to reach a location. This clearly shows that a distance metric that takes into account the transition function of the MDP is much more meaningful.[rev1]

To calculate a distance between two states we use the Generalised Graph Drawing Objective (GGDO) as presented by Wang et al. (2021). They present an improvement over the approach which was first investigated by Wu et al. (2018). These methods aim to learn an approximation of the Laplacian representation of an MDP. We have to use an approximation of the Laplacian representation because the exact Laplacian representation is very challenging to compute and this would limit us to environments with small state-spaces (Wang et al., 2021). When we consider the states and transitions of the MDP as nodes and edges in a graph, the Laplacian representation ($\phi$) consists of the $d$ smallest eigenvectors ($\vec{v}_1, \ldots, \vec{v}_d$) of the graph Laplacian.

$$\phi(s) = [\vec{v}_1[s], \ldots, \vec{v}_d[s]] \tag{6}$$

Laplacian representations contain geometry information of the state-transition graph that allows us to calculate the distance between two states taking into account the transition function of the MDP. The Laplacian representation of two connected states in this graph will be similar while the Laplacian representation of two states far away from each other in this graph will be dissimilar.[rev1]

Equation 7 describes the loss function used to learn the representation model as proposed by Wang et al. (2021). It consists of two main parts. The first term describes an attractive objective. It aims to bring the representation of consecutive states closer to each other. Here, $c$ is a weight vector with $c_1 = d$, $c_2 = d - 1$, $\ldots$, $c_d = 1$. In this equation "$\odot$" is used to describe a Hadamard or element-wise product. The remaining terms define a repulsive objective, aiming to push the representation of the current state away from a state ($\tilde{s}$) randomly sampled from the buffer. The second term is used to orthogonalise the representation and the third and fourth term regularize the representations away from zero.

$$\mathcal{L} = \mathbb{E}_{s_t, s_{t+1}, \tilde{s}} [\overbrace{||c \odot (\phi(s_t) - \phi(s_{t+1}))||^2}^{Attractive} + \overbrace{(\phi(s_t)^T \phi(\tilde{s}))^2 - ||\phi(s_t)||^2 - ||\phi(\tilde{s})||^2}^{Repulsive}] \tag{7}$$

Given that this loss function explicitly moves the representation of certain experienced states closer to each other and further away from other experienced states, the distribution of the data that is used for this training is important. We want the probability of the occurrence of states to be influenced only by their probability within the MDP. Therefore, we use a separate replay buffer that consists only of state transitions that are the result of random actions. This way, the policy of the agents minimally influences the Laplacian representation.

Using the learned state representation, we can then calculate a measure of the distance on our MDP graph between two states (Mahadevan, 2005). Wang et al. (2021) suggest using the learned Laplacian representation to learn a goal reaching policy using the following reward function , with $s_{goal}$ describing the state of the goal[rev1]:

$$r = -||\phi(s_{t+1}) - \phi(s_{goal})||^2 \tag{8}$$

This provides a reward that encourages going toward the goal state. We would like a reward that encourages going away from the initial state. Therefore, we propose the following reward function using the Laplacian representation of the initial state ($s_0$) and the next state ($s_{t+1}$):

$$r_i = ||\phi(s_{t+1}) - \phi(s_0)||^2 \tag{9}$$

In environments where the full state is not available, we use the joint observation to determine the Laplacian representation and the intrinsic reward. Using the joint observation will encourage the agents to explore the joint state-space, as shown by Toquebiau et al. (2024). However, in environments with a large number of agents there may be scalability issues when using the joint observation. In this case, an alternative approach may be required.[rev1] By calculating an individual intrinsic reward through an individual Laplacian representation, the scalability can be improved. However, this would reduce the ability of the agents to coordinate their exploration as shown by Toquebiau et al. (2024). Therefore, there will always be a trade-off between the scalability of the reward calculation and the ability of the agents to jointly explore the state-action space.[rev2]

$$r_i = ||\phi(o_{t+1}) - \phi(o_0)||^2 \tag{10}$$

This novel reward formulation will encourage the agents to move further away from the initial state, therefore providing motivation to the agents to overcome bottlenecks in the state-space.

## 5 Experiments

We will now dive deeper into the details of the experiments we perform in this work. First, we explain each of the environments we use. Next, we elaborate on some of the choices we made for our experimental setup.

### 5.1 `rel_overgen` Environment

The `rel_overgen` environment was presented by Toquebiau et al. (2024). In the environment, there are two agents. They can navigate across the reward function displayed in Figure 4, each controlling the position on one axis. The observations of the agents consists of a one-hot representation of the position on their respective axis. They do not observe the position on the axis controlled by the other agent. Episodes have a constant length of 40 timesteps, which is enough for the agents to be able to reach any position on the reward curve regardless of the starting position.[rev1] Each timestep, the agents choose to move one position in either direction on this axis or stay in the same position. The combined position of both agents gives an $(x, y)$

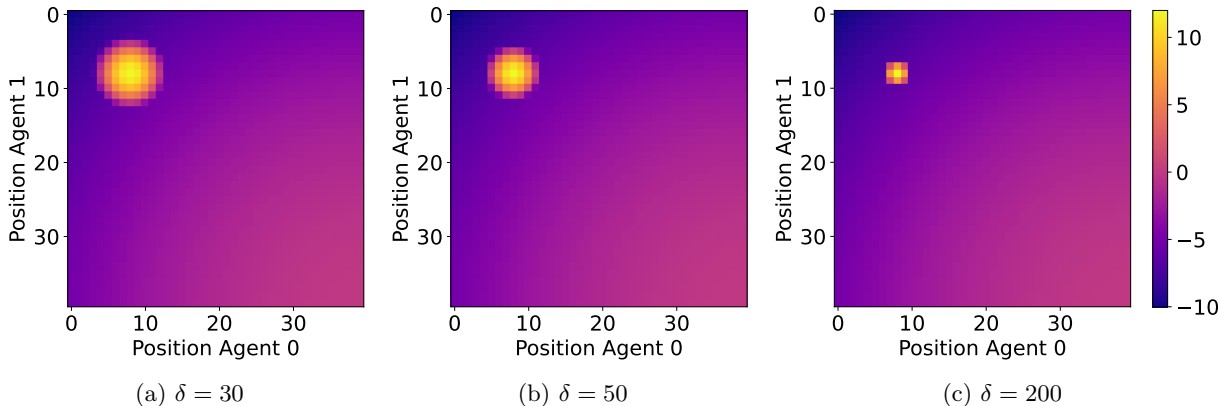

(a) $\delta = 30$            (b) $\delta = 50$            (c) $\delta = 200$

Figure 4: Reward function for `rel_overgen` environment. The $\delta$ parameter controls the width of the optimal reward peak. A higher value of $\delta$ makes it more difficult for the agents to learn a policy where they effectively reach the optimal reward peak.[rev1]

Table 1: Meaning and values for the reward function parameters in the `rel_overgen` environment.

| Parameter | Value | Comment |
|:---------:|:-----:|:--------|
| $R^+$ | 12 | Value of the optimal reward peak |
| $R^-$ | 0 | Value of the sub-optimal reward peak |
| $r_x^+$ | 8 | x-coordinate of the optimal reward peak |
| $r_y^+$ | 8 | y-coordinate of the optimal reward peak |
| $r_x^-$ | 40 | x-coordinate of the sub-optimal reward peak |
| $r_y^-$ | 40 | y-coordinate of the sub-optimal reward peak |
| $D$ | 40 | Number of positions on each axis |
| $\delta$ | 200 | Width of the optimal reward peak |

position on the reward curve which[rev1] is used to calculate the reward. We use the same reward function as presented by Toquebiau et al. (2024) in Equation 11 and visualised in Figure 4. Table 1 shows the value for each of the parameters in Equation 11. In their work, Toquebiau et al. (2024) test three configurations for $\delta$, $\delta = 30$, $\delta = 40$ and $\delta = 50$. We make the environment more challenging by increasing the $\delta$-value to 200.

$$r_i(x, y; \delta) = \max\Big(R^+ - \frac{\delta}{D}\big[(x - r_x^+)^2 + (y - r_y^+)^2\big],$$
$$R^- - \frac{1}{8D}\big[(x - r_x^-)^2 + (y - r_y^-)^2\big]\Big). \tag{11}$$

The `rel_overgen` environment is designed to investigate how well methods can deal with relative overgeneralisation. In addition to this, the agents also have high interdependence and need perfect coordination to reach the maximal possible return in the environment.

## 5.2 Laser Learning Environment

The Laser Learning Environment (LLE) was introduced by Molinghen et al. (2023) as an environment ideally suited to investigate challenges in multi-agent systems not yet present in other MARL benchmarks , specifically interdependence, perfect coordination and zero-incentive dynamics. [rev1]

In the LLE, there is a set of agents that need to collect gems and reach one of the end tiles. The difficulty in this environment is the lasers that are placed in the environment as can be seen in Figure 5. Only an agent with the colour matching the laser can pass through the laser and block it. Therefore, agents have to

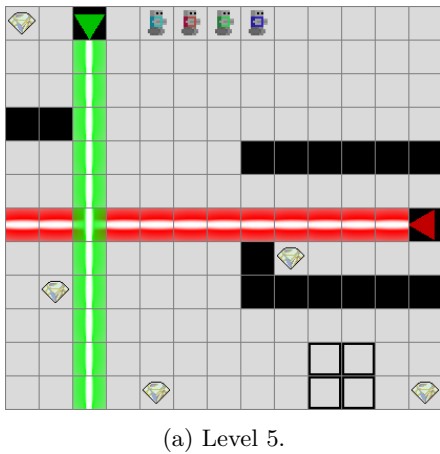
(a) Level 5.

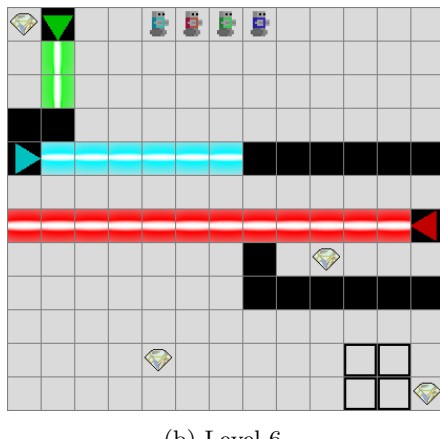
(b) Level 6.

Figure 5: Laser Learning Environment (LLE). The agents need to reach the exit tiles, while avoiding the lasers belonging to other agents. In order to succeed together, the agents have to block their laser for other agents to be able to pass them. The agents receive an additional reward for picking up the gems.[rev1]

coordinate their behaviour to block the lasers when other agents need to pass in order for all agents to be able to reach an end tile.

Agents are rewarded with +1 for collecting gems and for each agent reaching an end tile. If all agents reach an end tile, the agents are rewarded by an additional +1. The agents have five possible actions (up, down, left, right, stay). The environment is fully observable, providing the agents with a layered observation indicating the location of each of the environment elements (agents, lasers, gems, walls and end tiles). Figure 18 in Appendix A.2 shows the different observation layers.[rev1]

The LLE is designed to test how well methods can handle multi-agent exploration challenges. In their work, Molinghen et al. (2023) focus on interdependence, perfect coordination and zero-incentive dynamics as explained in Section 1. The way the lasers work creates a very challenging scenario for the agents. They must learn to block the correct laser at the correct time, so the necessary agents can pass through. There is no reward for agents passing the laser which makes it hard to discover the need for this behaviour.

## 5.3 Experimental setup

For most hyperparameters, we follow the values proposed by Molinghen et al. (2023) and Toquebiau et al. (2024). We made a few modifications where we observed better results using other values. Unless otherwise specified, each of the the experiments uses JRND as intrinsic reward, $l = 10$ for the LLE and $l = 20$ for the `rel_overgen` environment as policy sampling period (see Section 6.2) and the mixed configuration for synchronised exploration (see Section 6.3). A complete description of the model architectures and hyperparameters used in our experiments can be found in Appendix A.3 and A.4. For our experiments, we use RLlib (Liang et al., 2018) and Tune (Liaw et al., 2018) inside the Ray framework (Moritz et al., 2017).

The implementation of our agents is based on the VDN/QMIX implementation in RLlib (Liang et al., 2018) which is based on the implementation in PyMARL (Samvelyan et al., 2019).

For each experiment, we perform five runs. All the agents use parameter sharing and use only the current observation, no observation history. We show the average results throughout training along with the 95% confidence interval. In a second graph, we show the distribution of the performance of the different runs in a violin graph calculated on the results of the last 100 training iterations. The violin plot also shows the minimum, maximum and median performance of each of the configurations.

Table 2: Overview of the different tested architectures.

| Abbreviation | Single / Dual-Policy | Exploration Approach |
|---|---|---|
| $\epsilon$G-DP | Dual | $\epsilon$-greedy |
| WR-SP | Single | Weighted Rewards |
| WQ-DP (ours) | Dual | Weighted Q-values |
| $\epsilon$S-DP (ours) | Dual | $\epsilon$-sampled |

Table 3: Comparison of the average return during the last 10% of training using different agent architectures.

| | rel_overgen | LLE Level 5 | LLE Level 6 |
|---|---|---|---|
| $\epsilon$G-DP | $-33.14 \pm 2.20$ | $3.55 \pm 0.66$ | $2.74 \pm 1.74$ |
| $\epsilon$G-DP (FO) | $215.17 \pm 10.43$ | N/A | N/A |
| WR-SP | $17.62 \pm 103.13$ | $4.20 \pm 0.60$ | $3.40 \pm 1.67$ |
| WQ-DP | $129.85 \pm 78.95$ | $5.95 \pm 1.51$ | $3.38 \pm 0.26$ |
| $\epsilon$S-DP | $181.56 \pm 89.24$ | $7.35 \pm 0.88$ | $5.38 \pm 0.16$ |

## 6 Results

In this section, we look at the results of our experiments and analyse the different aspects of designing our exploration agent.

### 6.1 Agent Architecture

In our first experiment, we compare the architectures that were presented in Section 4.1. We also compare them with two state-of-the-art baselines. $\epsilon$G-DP is a basic approach but is still widely used in MARL. WR-SP is the approach that the majority of learned exploration methods employ. We did not compare with the approaches presented by Liu et al. (2021) and Böhmer et al. (2019) because this would introduce too many confounding variables. The focus of this experiment is to compare the use of dual-policy architectures with the currently most used agent architectures while keeping all other elements of the agents the same.

We use JRND, which closely resembles JIM (Toquebiau et al., 2024), a state-of-the-art method in multi-agent exploration, specifically focused on dealing with relative overgeneralisation. We chose to use JRND instead of the more advanced JIM approach because we want to focus on the performance influence of the different architectures without adding too much complexity with the design of the intrinsic reward. However, the proposed agent architectures do not rely on a specific intrinsic reward (see Section 6.4) and can therefore be used along with JIM or other intrinsic reward approaches.

In Table 2, we provide an overview of the different architectures and their properties. Figure 6, 7 and 8 and Table 3 show the results. In all the experiments, we see that the architecture makes a clear difference in the achieved return of the agents. In each of the environments, we see that $\epsilon$S-DP clearly outperforms both $\epsilon$G-DP and WR-SP. WQ-DP performs better than $\epsilon$G-DP and WR-SP, but we see that it does not perform as well as $\epsilon$S-DP. In level 5 of the LLE, we see that WQ-DP no longer consistently outperforms WR-SP. It achieves approximately the same average return but has a smaller spread.

In our experiments on the rel_overgen environment, we also included an experiment where we use the $\epsilon$G-DP architecture, but we modify the environment to be fully observable ($\epsilon$G-DP (FO)). This means that we added the position along the axis of the other agent, as well as the current timestep. With this additional experiment, we show that the partial observability is the factor that makes it hard for the agents to explore this environment and find an effective policy. This is a different challenge as the LLE which is fully observable but challenging due to the state-bottlenecks introduced by the lasers. However, in both these environments, using a dual-policy architecture, especially $\epsilon$S-DP, makes a significant performance difference.

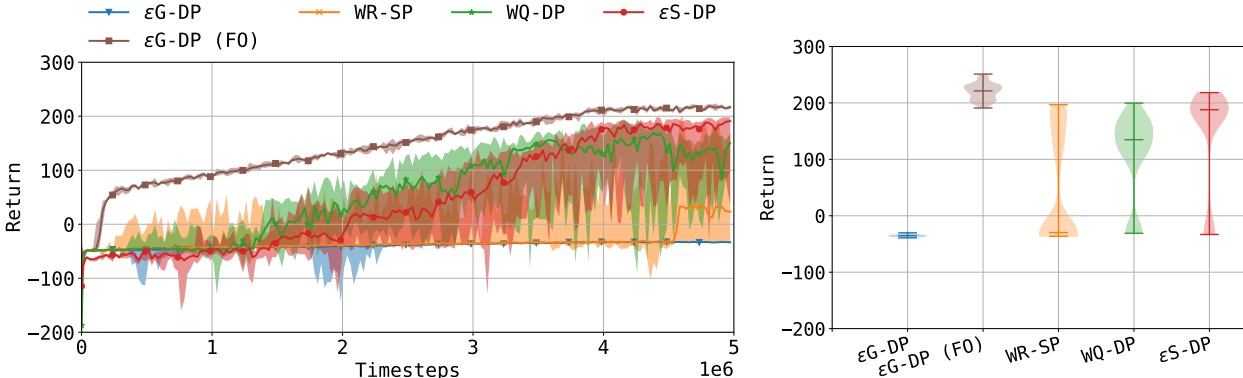

Figure 6: Comparison of the return achieved in the `rel_overgen` environment using different agent architectures. $\epsilon$S-DP (FO) describes an additional experiment where we make the environment fully observable by adding the location along the axis controlled by the other agent and the timestep to the observation.[rev1]

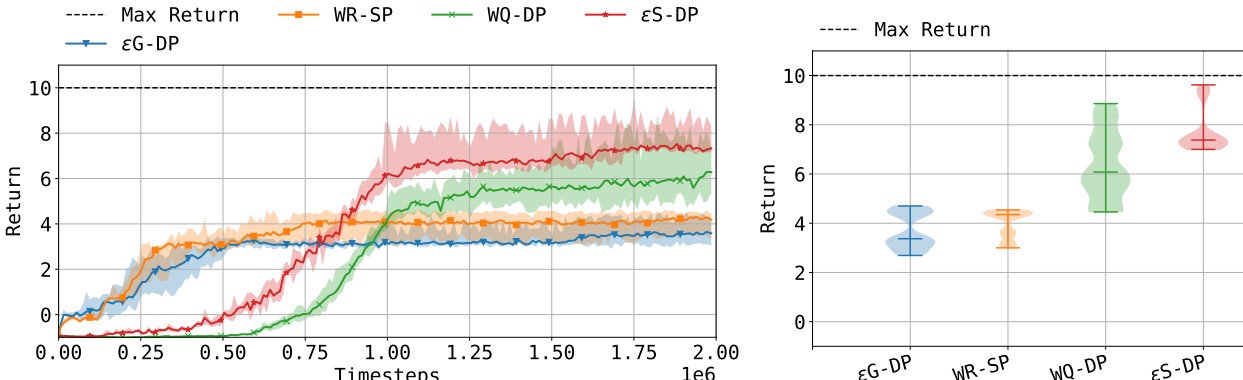

Figure 7: Comparison of the return achieved in level 5 of the LLE using different agent architectures.

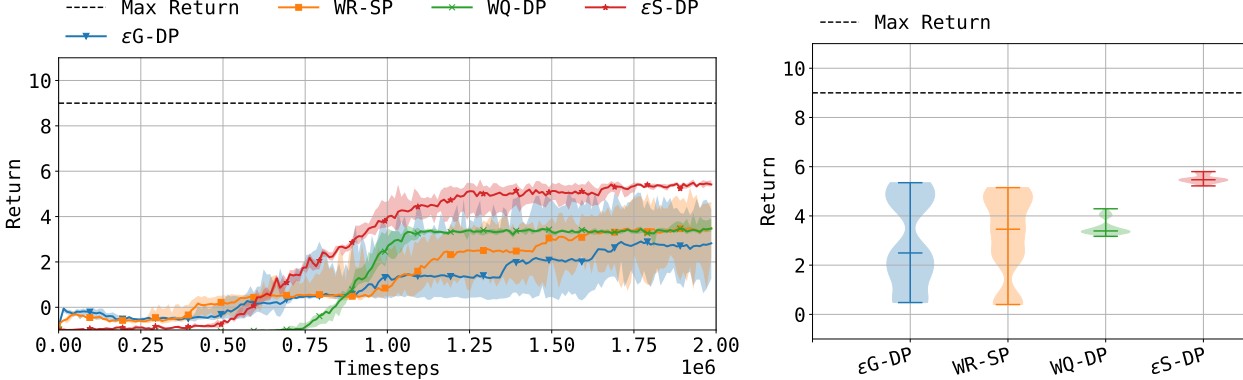

Figure 8: Comparison of the return achieved in level 6 of the LLE using different agent architectures.

## 6.2 Policy Sampling Period

When using $\epsilon$S-DP, we have more control over the exploration. A first example of this is that we can control how often we re-sample which policy to use to select an action. We hypothesise that exploring / exploiting over longer periods instead of re-sampling each timestep can be beneficial in training. Something similar has also been done for single agent RL in the works of Dabney et al. (2020) and Bagot et al. (2020). Dabney et al.

(2020) propose temporally-extended $\epsilon$-greedy where the randomly sampled action is repeated for a random duration instead of only one timestep. This approach was shown to be very effective in their experiments. Bagot et al. (2020) show that allowing the exploration policy to remain in control for a longer period each time the exploration option is chosen can significantly improve the performance of the agent.[rev1]

Figure 9, 10 and 11 and Table 4 show the results of a comparison between a variety of values for the policy sampling period $l$ ranging from re-sampling each timestep ($l = 1$) to sampling only once each episode. In the results, we see that, in each of the environments, the agents clearly benefit from using the policies over longer periods. For the LLE, the optimal configuration is $l = 10$ for both levels. In the `rel_overgen` environment, the optimal configuration is $l = 20$. Further increasing the policy sampling period no longer provides a performance benefit.

We can thus conclude that the ability to control the policy sampling period of the $\epsilon$S-DP architecture is a great advantage compared to the other architectures. We clearly see that using the policies over a longer period is a big contributing factor to the superior performance of $\epsilon$S-DP compared to the other architectures evaluated in Section 6.1.

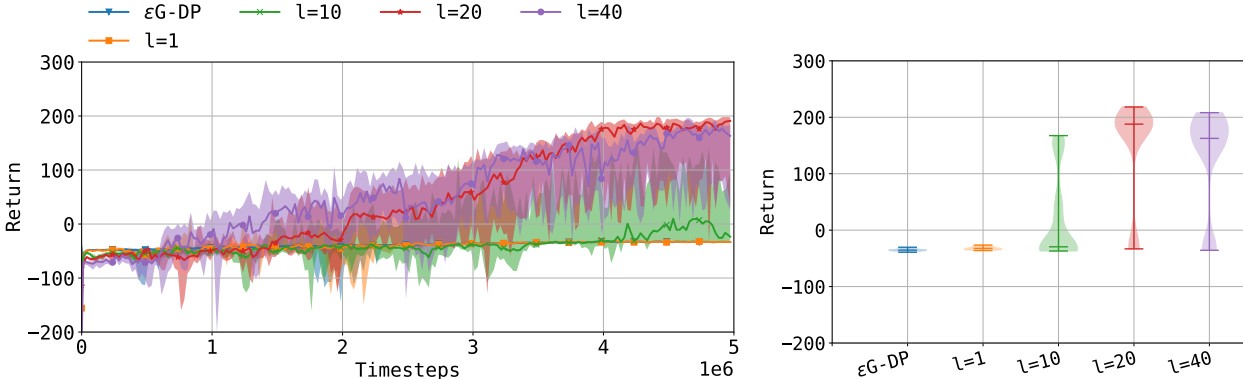

Figure 9: Comparison of the return achieved in the `rel_overgen` environment using different sampling frequencies.

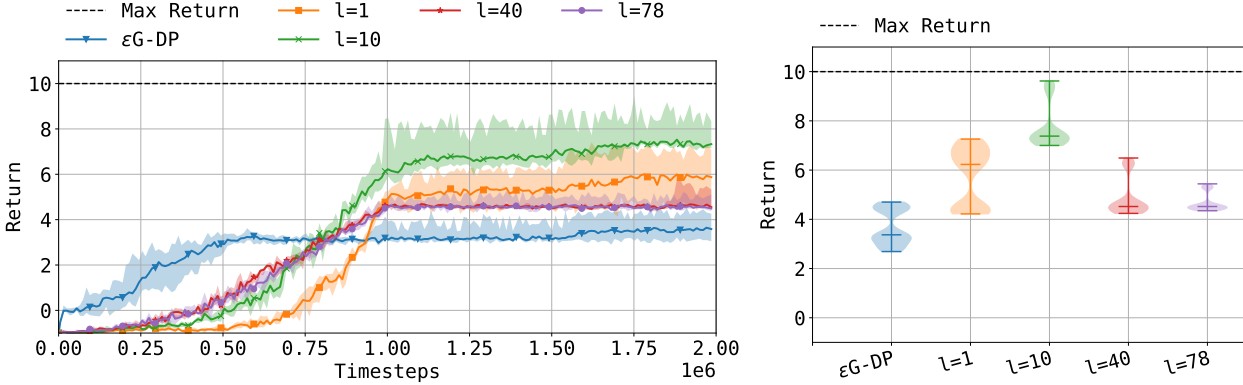

Figure 10: Comparison of the return achieved in level 5 of the LLE using different sampling frequencies.

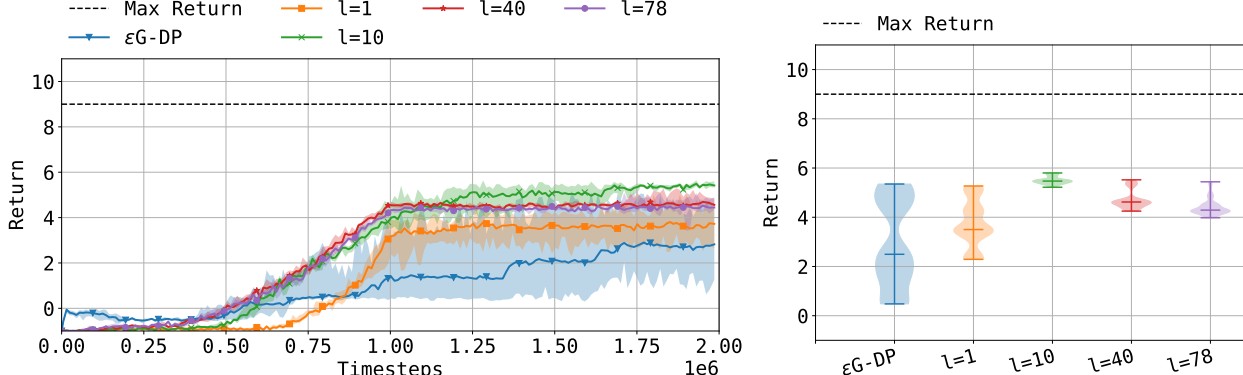

Figure 11: Comparison of the return achieved in level 6 of the LLE using different sampling frequencies.

Table 4: Comparison of the average return during the last 10% of training using different sampling frequencies.

| | rel_overgen | | | LLE Level 5 | LLE Level 6 |
|---|---|---|---|---|---|
| $\epsilon$G-DP | $-33.14 \pm 2.20$ | | $\epsilon$G-DP | $3.55 \pm 0.66$ | $2.74 \pm 1.74$ |
| $l = 1$ | $-33.11 \pm 2.18$ | | $l = 1$ | $5.86 \pm 1.22$ | $3.65 \pm 0.93$ |
| $l = 10$ | $-5.45 \pm 83.04$ | | $l = 10$ | $7.35 \pm 0.88$ | $5.38 \pm 0.16$ |
| $l = 20$ | $181.56 \pm 89.24$ | | $l = 40$ | $4.60 \pm 0.41$ | $4.59 \pm 0.38$ |
| $l = 40$ | $169.54 \pm 84.89$ | | $l = 78$ | $4.54 \pm 0.36$ | $4.45 \pm 0.42$ |

## 6.3 Synchronised Exploration

Another aspect of exploration that can be controlled when using $\epsilon$S-DP is whether the agents explore simultaneously or not. Within the $\epsilon$S-DP architecture, there are two level through which we can control synchronised exploration. First, on the macro-level, we can synchronise the sampling for either the exploration or exploitation policy. Second, on the micro-level, we can synchronise whether, if using the exploration policy, we sample a random action or an action from the exploration policy. We test three different synchronisation configurations as can be seen in Table 5. The other experiments in this paper use the mixed configuration.[rev1]

Table 5: Different configurations of simultaneous exploration.

| Configuration | Macro Synchronisation | Micro Synchronisation |
|---|---|---|
| Independent | ✗ | ✗ |
| Mixed | ✗ | ✓ |
| Synchronised | ✓ | ✓ |

Figure 12, 13 and 14 and Table 6 show the results of comparing different levels of synchronised exploration. We see the biggest difference between the configurations in the results for level 5 of the LLE. Here, we see that the mixed configuration achieves the most consistent and highest return. The synchronised configuration consistently has a lower return and the independent configuration achieves a very wide range of returns. In the other experiments, the difference between the configurations is very minor. Therefore, we can conclude that synchronising the exploration across the different agents only has a small influence on the achieved return.

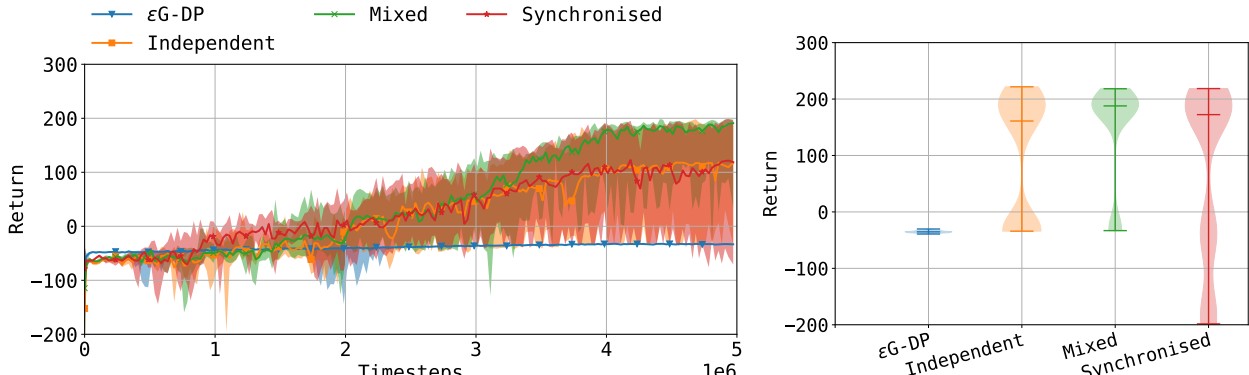

Figure 12: Comparison of different levels of simultaneous exploration in the `rel_overgen` environment.

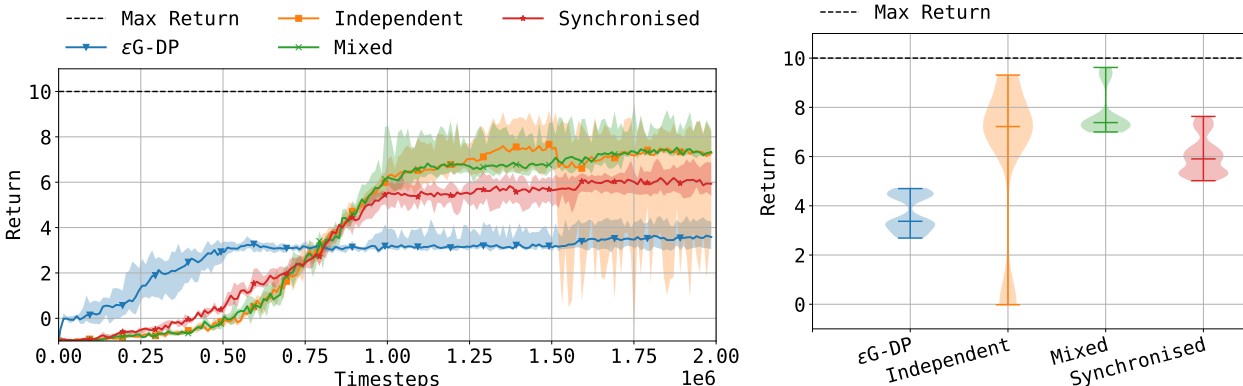

Figure 13: Comparison of different levels of simultaneous exploration in level 5 of the LLE.

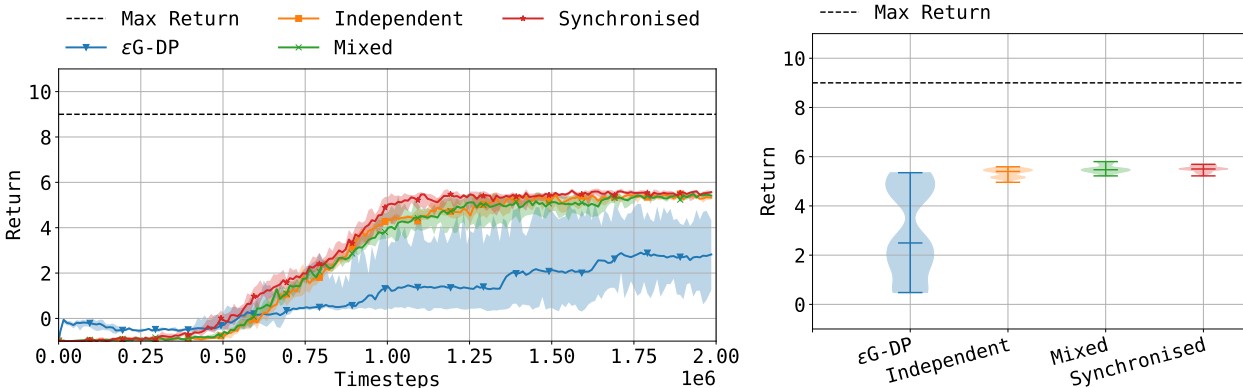

Figure 14: Comparison of different levels of simultaneous exploration in level 6 of the LLE.

Table 6: Comparison of the average return during the last 10% of training using different levels of simultaneous exploration.

|  | `rel_overgen` | LLE Level 5 | LLE Level 6 |
|---|---|---|---|
| $\epsilon$G-DP | $-33.14 \pm 2.20$ | $3.55 \pm 0.66$ | $2.74 \pm 1.74$ |
| Independent | $113.49 \pm 107.78$ | $7.33 \pm 3.38$ | $5.37 \pm 0.17$ |
| Mixed | $181.56 \pm 89.24$ | $7.35 \pm 0.88$ | $5.38 \pm 0.16$ |
| Synchronised | $104.24 \pm 119.33$ | $6.03 \pm 0.68$ | $5.51 \pm 0.16$ |

### 6.4 Intrinsic Reward

Another aspect that influences the exploration of the agents is the intrinsic reward that is used. In Section 4.2, we presented the Laplacian Representation Reward (LRR). We compare this with JRND, which is based on an approach that has been well proven in single-agent exploration. Table 7 and Figure 15, 16 and 17 show the results. For this experiment, we used the $\epsilon$S-DP architecture (except for the baseline $\epsilon$G-DP configuration). We see that the difference in return resulting from the different reward functions is much smaller than the effect of the agent architecture. However, overall we can see that LRR results in a larger spread in the achieved return across the different experiments. LRR clearly outperforms the baseline $\epsilon$G-DP, showing the potential of using Laplacian representations to calculate an intrinsic reward. In the LLE, we see that the agents using LRR learn slightly faster, which could be beneficial in settings where the number of allowed interactions with the environment is limited. We do not see this in the `rel_overgen` environment. We hypothesize that this is because in the LLE the biggest challenge to overcome with the exploration strategy is the state-bottlenecks caused by the lasers. The agents using LRR are encouraged to move further away from the initial position and are therefore rewarded for overcoming the state-bottlenecks. In the `rel_overgen` environment, the challenging part is coordinating between the two agents even though the environment is partially observable. Consequently, we see that in this environment LRR is not as effective as JRND. We can conclude that LRR shows potential to aid in overcoming bottlenecks in the state-space. In addition, these experiments show that $\epsilon$S-DP does not rely on the use of a specific intrinsic reward to be effective.

An important note here is that for each of the runs we used the exact same hyperparameters between the runs using JRND and the runs using LRR (except for the reward calculation configurations). However, in the `rel_overgen` environment, we use a different value for the policy sampling period. For JRND, we saw in Section 6.2 that $l = 20$ results in the best return. For LRR, we discovered during our hyperparameter search that $l = 10$ is best. Therefore, we used $l = 10$ for the experiments using LRR.

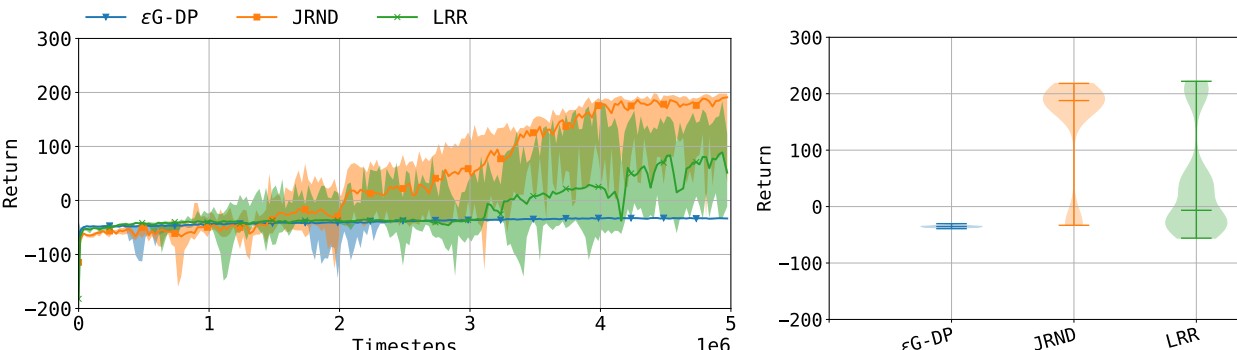

Figure 15: Comparison of the return achieved in the `rel_overgen` environment using different intrinsic rewards.

Table 7: Comparison of the average return during the last 10% of training using different intrinsic rewards.

|  | `rel_overgen` | LLE Level 5 | LLE Level 6 |
|---|---|---|---|
| $\epsilon$G-DP | $-33.14 \pm 2.20$ | $3.55 \pm 0.66$ | $2.74 \pm 1.74$ |
| JRND | $181.56 \pm 89.24$ | $7.35 \pm 0.88$ | $5.38 \pm 0.16$ |
| LRR | $63.75 \pm 89.72$ | $7.43 \pm 1.78$ | $5.38 \pm 0.46$ |

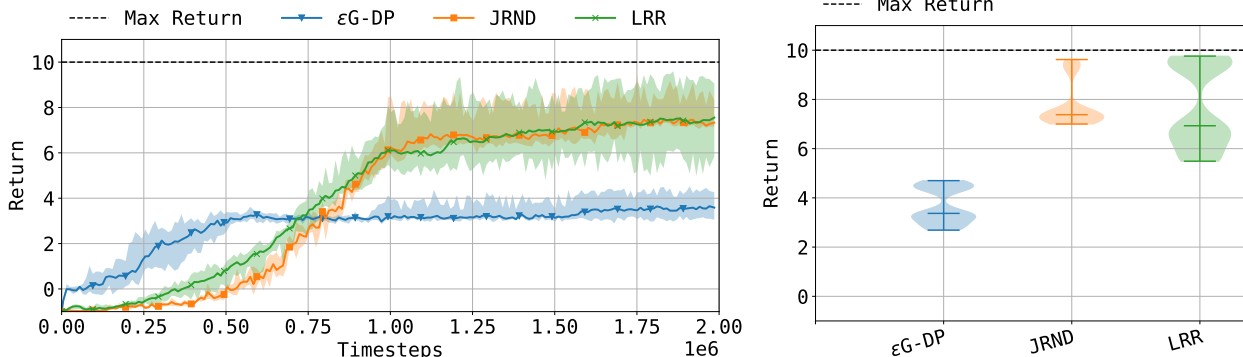

Figure 16: Comparison of the return achieved in level 5 of the LLE using different intrinsic rewards.

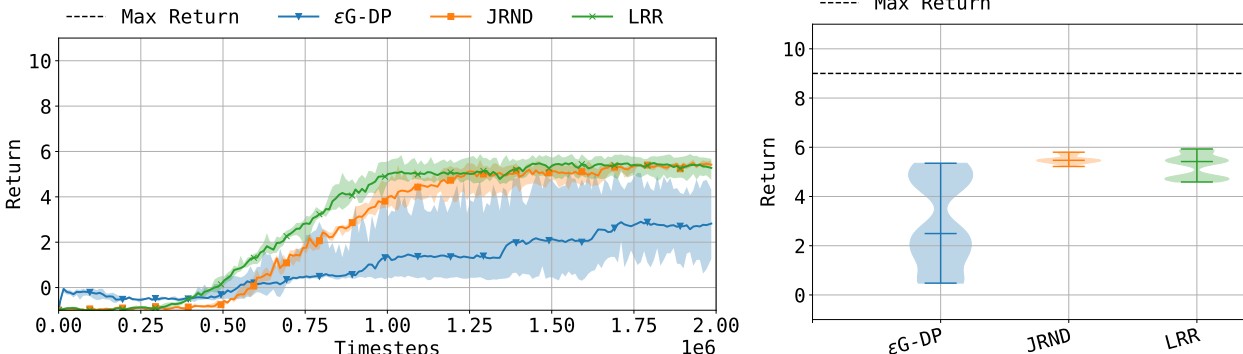

Figure 17: Comparison of the return achieved in level 6 of the LLE using different intrinsic rewards.

# 7 Conclusion & Future Work

In this work, we investigate the use of dual-policy architectures for exploration in MARL. We present two possible implementations of the dual-policy architecture, Weighted-Q Dual-Policy (WQ-DP) and $\epsilon$-Sampled Dual-Policy ($\epsilon$S-DP). We compare these with with the most commonly used state-of-the-art baselines, $\epsilon$-Greedy Dual-Policy ($\epsilon$G-DP) and Weighted-Rewards Single-Policy (WR-SP). Our results clearly show that both dual-policy approaches outperform these baselines. $\epsilon$S-DP performs best out of the two implementations.

We evaluate on two challenging environments that exhibit many exploration challenges. Our results show that through the use of our architectures, the exploration policies are able to perform better and deal with these challenges more effectively. By separating exploration and exploitation, both policies are able to focus on a single task and this allows them to learn more successful behaviours.[rev1]

In further experiments, we analyse which aspects make $\epsilon$S-DP perform better than the other approaches we tested. The first two aspects we investigate are configurations unique to our most successful architecture, $\epsilon$S-DP. First, we look at whether the policy sampling frequency has a big influence on the return. We see that being able to control this parameter enables the agents to achieve a much higher return. Next, synchronizing the choice for exploration or exploitation between the different agents does not show to have as much influence on the performance of the agents. Finally, we look at an alternative to the used intrinsic reward (JRND) based on Laplacian representations (LRR). Overall, we see that they achieve very similar results, although JRND provides more consistent results than LRR. However, the results show the potential of using Laplacian representations to calculate an intrinsic reward in environments that contain bottlenecks in the state-space such as the LLE[rev1]. In addition, these results show that $\epsilon$S-DP remains effective when using a different intrinsic reward approach.

In this research area, there are many paths for future research. In this work, we analysed the influence of two of the introduced hyperparameters, the policy sampling period and the exploration synchronisation. However, an additional parameter that can be investigated is the value of $\epsilon_{macro}$ and its schedule.[rev2] The presented LRR showed potential in our results, but it could still be improved. Gomez et al. (2023) present a novel approach to learning a Laplacian representation that looks very promising. Using this method in our reward calculations could improve the learned Laplacian representation and therefore also give a more accurate reward signal to the agents. We can also further look into leveraging the information contained within the individual features of the Laplacian representation. For example, by using a weighted sum of the features where the weights change throughout training instead of a norm. This would make the exploration behaviour more diverse. In addition, since LRR learns a representation of the state where states close together in the MDP graph will have similar representation, we hypothesize that when using LRR the agents will suffer less from the Noisy-TV problem. In future, work this should be further investigated. In the related work we saw many methods that use both a novelty reward and an episodic reward. Including an episodic term to our intrinsic reward calculation could further improve our results. Finally, in this work we presented two possible architectures to combine the exploitation and exploration policies. However, in future work it would be interesting to investigate an intelligent way to determine when to use which policy. Bagot et al. (2020) have already presented a successful method for this in single-agent RL.

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

## A Appendix

### A.1 Synchronised Exploration Details[rev1]

---

**Algorithm 6** Policy Mixing: $\epsilon$-Sampled Dual-Policy (independent configuration (See section 6.3)[rev1])

---

$\epsilon_{macro} \in [0, 1]$
$\epsilon_{micro} \in [0, 1]$
$l \leftarrow$ policy sampling period (see Section 6.2)
$i \leftarrow l$
$choice^a \leftarrow$ "" for $a \in \mathbb{A}$
**function** POLICYMIXING($\overrightarrow{Q}_e$, $\overrightarrow{Q}_i$, $\mathbb{U}(s_t)$)
    $i \leftarrow i + 1$
    **for** $a \in \mathbb{A}$ **do**
        **if** $i > l$ **then**
            $choice^a \leftarrow \begin{cases} "exploitation" & \text{with probability of } (1 - \epsilon_{macro}) \\ "exploration" & \text{with probability of } \epsilon_{macro} \end{cases}$
        **end if**
        $u_t^a \leftarrow \begin{cases} \underset{u'^a}{\arg\max}(\overrightarrow{Q}_e^a) & \text{if } choice^a == "exploitation" \\ \underset{u'^a}{\arg\max}(\overrightarrow{Q}_i^a) & \text{if } choice^a == "exploration" \text{ and with probability of } (1 - \epsilon_{micro}) \\ \sim U(\mathbb{U}^a(s_t)) & \text{if } choice^a == "exploration" \text{ and with probability of } \epsilon_{micro} \end{cases}$
    **end for**
    **if** $i > l$ **then**
        $i \leftarrow 0$
    **end if**
    $u_t \leftarrow [u_t^1, \ldots, u_t^N]$
    **return** $u_t$
**end function**

---

---

**Algorithm 7** Policy Mixing: $\epsilon$-Sampled Dual-Policy (synchronised configuration (See section 6.3))rev1

---

$\epsilon_{macro} \in [0, 1]$
$\epsilon_{micro} \in [0, 1]$
$l \leftarrow$ policy sampling period (see Section 6.2)
$i \leftarrow l$
$choice \leftarrow$ ""
**function** POLICYMIXING($\overrightarrow{Q}_e$, $\overrightarrow{Q}_i$, $\mathbb{U}(s_t)$)
    $i \leftarrow i + 1$
    $choice\_micro \leftarrow \begin{cases} \text{"policy"} & \text{with probability of } (1 - \epsilon_{micro}) \\ \text{"random"} & \text{with probability of } \epsilon_{micro} \end{cases}$
    **if** $i > l$ **then**
        $i \leftarrow 0$
        $choice \leftarrow \begin{cases} \text{"exploitation"} & \text{with probability of } (1 - \epsilon_{macro}) \\ \text{"exploration"} & \text{with probability of } \epsilon_{macro} \end{cases}$
    **end if**
    **for** $a \in \mathbb{A}$ **do**
        $u_t^a \leftarrow \begin{cases} \underset{u'^a}{\text{argmax}}(\overrightarrow{Q}_e^a) & \text{if } choice == \text{"exploitation"} \\ \underset{u'^a}{\text{argmax}}(\overrightarrow{Q}_i^a) & \text{if } choice == \text{"exploration"} \text{ and if } choice\_micro == \text{"policy"} \\ \sim U(\mathbb{U}^a(s_t)) & \text{if } choice == \text{"exploration"} \text{ and if } choice\_micro == \text{"random"} \end{cases}$
    **end for**
    $u_t \leftarrow [u_t^1, \ldots, u_t^N]$
    **return** $u_t$
**end function**

---

## A.2 Further Details of the Laser Learning Environment[rev1]

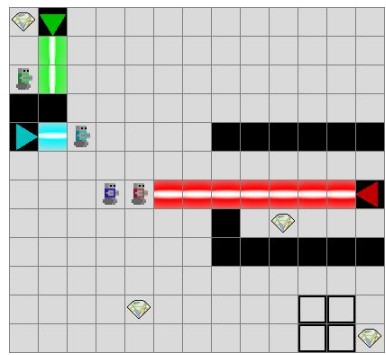

(a) Environment State

(b) Agent 0     (c) Agent 1     (d) Agent 2     (e) Agent 3

(f) Laser 0     (g) Laser 1     (h) Laser 2     (i) Laser 3

(j) Walls     (k) Voids     (l) Gems     (m) Exits

Figure 18: Layered observation of the Laser Learning Environment (LLE).[rev1]

### A.3 Model Architecture

In this section, we summarize the model architectures we use in our experiments in each of the environments. We use the same architecture for both the exploitation and exploration policies.

#### A.3.1 LLE

| Layer Type | Activation | Output Size | Stride | Kernel |
|---|---|---|---|---|
| Input | | $(13 \times 12 \times 13)^{\mathbf{rev1}}$ | | |
| Conv2D | ReLU | $(32 \times 10 \times 11)$ | 1 | $(3 \times 3)$ |
| Conv2D | ReLU | $(64 \times 8 \times 9)$ | 1 | $(3 \times 3)$ |
| Conv2D | ReLU | $(32 \times 6 \times 7)$ | 1 | $(3 \times 3)$ |
| Flatten | | 1344 | | |
| Concat | | 1348 | | |
| Linear | ReLU | 64 | | |
| Linear | ReLU | 64 | | |
| Linear | | 5 | | |

#### A.3.2 `rel_overgen`

| Layer Type | Activation | Output Size |
|---|---|---|
| Input | | 40 |
| Linear | ReLU | 64 |
| GRU | ReLU | 64 |
| Linear | | 3 |

### A.4 Hyperparameters

In this section, we give an overview of the used hyperparameters. We have reached these hyperparameters starting from the hyperparameters used by Molinghen et al. (2023) for the LLE and by Toquebiau et al. (2024) for the `rel_overgen` environment. We further improved and expanded them for our experiments and architectures using grid searches.

#### A.4.1 Exploitation Policy

| Parameter | Value (LLE) | Value (`rel_overgen`) |
|---|---|---|
| Value Factorisation | VDN | QMIX |
| Learning Rate | 0.0005 | 0.0007 |
| Discount Factor | 0.95 | 0.99 |
| Soft Update Rate | 0.01 | 0.005 |
| Batch Size | 512 | 1280 |
| Replay Buffer Size | 50e3 | 5e3 |
| Training Interval | 40 | 40 |

#### A.4.2 Exploration Policy

| Parameter | Value (LLE) | Value (`rel_overgen`) |
|---|---|---|
| Value Factorisation | VDN | QMIX |
| Learning Rate | 0.005 | 0.005 |
| Discount Factor | 0.9 | 0.9 |
| Soft Update Rate | 0.01 | 0.01 |
| Batch Size | 512 | 1280 |
| Replay Buffer Size | 50e3 | 5e3 |
| Training Interval | 40 | 40 |

### A.4.3 $\epsilon$-Greedy Dual-Policy

| Parameter | Value (LLE) | Value (`rel_overgen`) |
|---|---|---|
| $\epsilon$ | $1 \to 0.05$ linearly over 1M timesteps | $0.3 \to 0.05$ linearly over 4M timesteps |

### A.4.4 Weighted-Rewards Single-Policy

| Parameter | Value (LLE) | Value (`rel_overgen`) |
|---|---|---|
| Intrinsic Reward Factor | $2 \to 0$ linearly over 1M timesteps | $2 \to 0$ linearly over 4M timesteps |
| $\epsilon$ | $1 \to 0.05$ linearly over 1M timesteps | $0.3 \to 0.05$ linearly over 4M timesteps |

### A.4.5 Weighted-Q Dual-Policy

| Parameter | Value (LLE) | Value (`rel_overgen`) |
|---|---|---|
| Exploration Q-value Factor | $2 \to 0.1$ linearly over 1M timesteps | $2 \to 0.1$ linearly over 4M timesteps |
| $\epsilon$ | $1 \to 0.05$ linearly over 1M timesteps | $0.3 \to 0.05$ linearly over 4M timesteps |

### A.4.6 $\epsilon$-Sampled Dual-Policy

| Parameter | Value (LLE) | Value (`rel_overgen`) |
|---|---|---|
| $\epsilon$ (Micro) | $1 \to 0.3$ linearly over 200k timesteps | $1 \to 0.3$ linearly over 500k timesteps |
| $\epsilon$ (Macro) | $1 \to 0.05$ linearly over 1M timesteps | $0.3 \to 0.05$ linearly over 4M timesteps |
| $l$ | 10 | 20 |
| Exploration Synchronisation | Mixed | Mixed |

### A.4.7 Random Network Distillation

| Parameter | Value (LLE) | Value (`rel_overgen`) |
|---|---|---|
| Learning Rate | 0.005 | 0.0005 |
| Batch Size | 512 | 1280 |
| Latent Size | 10 | 10 |
| Reward Clip | 5 | N/A |
| Hidden layers | [64, 64] | [16, 16] |
| Activation | ReLU | ReLU |

### A.4.8 Laplacian Representation Reward

| Parameter | Value (LLE) | Value (`rel_overgen`) |
|---|---|---|
| Learning Rate | 0.005 | 0.005 |
| Batch Size | 512 | 1280 |
| Latent Size | 10 | 10 |
| Hidden layers | [64, 64] | [16, 16] |
| Activation | ReLU | ReLU |
| Replay Buffer Size | 10e3 | 1e3 |

