# OpenReview forum: "Dual-Policy Architecture for Multi-Agent Exploration"
_TMLR — Rejected by TMLR_

### Review · Reviewer_7tWo · 2025-02-24

**Summary Of Contributions:**

The proposes new methods for multi-agent reinforcement learning (MARL) focusing on infusing exploration with a dual-policy approach. In all the proposed variations of the approach, one of the policies is trained on an exploitation reward and one of the policies is trained on an exploration reward. Based on this framework, the authors propose different ways of formulating an exploration reward, Joint Random
Network Distillation (JRND) and Laplacian Representation Reward (LRR), and two ways of choosing actions, summing Q-values in Weighted-Q Dual-Policy (WR-SP) and ϵ-Sampled Dual-Policy (ϵS-DP).

The paper starts with an introduction describing challenges related to exploration in reinforcement learning, which are further augmented in MARL settings. Next, the paper describes related work in Section 2 focusing on single-agent RL exploration and MARL exploration approaches. This is followed by Section 3 which outlines background before introducing the methods in Section 4. Section 4 describes the dual policy architectures, as well WR-SP and ϵS-DP followed by a description of LRR. Section 5 presents the main experiments focusing on two environments: rel_overgen and  Laser Learning Environment. The first set of results benchmark WR-SP and ϵS-DP against some MARL showing general outperformance. The second of results show an analysis of exploration parameters in ϵS-DP, which can be tuned for different task and environment settings. The third set of results focus on synchronizing exploration across multiple agents and the fourth set of results explores different intrinsic reward settings. Across all sets of results, the authors perform an analysis stating different options and the best-performing method for each setting. The paper finishes with a conclusion and a discussion of future work.

Overall, the proposed contributions are:
* A new method for MARL settings based on a dual-policy architecture with tunable exploration.
* An analysis of different exploration settings and intrinsic reward settings for the proposed ϵS-DP.

**Audience:**

Yes

**Broader Impact Concerns:**

No major broader impact concerns.

**Claims And Evidence:**

No

**Requested Changes:**

Requested Changes:
* Edit the paper to fit into the recommended 12 pages of content. Much of the background and part of the related work could be moved into an appendix, thereby focusing the main paper on the most important content.
* Provide more detailed motivation for the chosen set of environments, baselines and analysis questions. This will help make the better structure and main takeaways clearer. Currently all environments are cooperative environments; it would be good to have more detail on why those were chosen and how they compare contrast to competitive environments. Some examples of competitive environments are predator-prey style environments which can be found in [1].
* Provide more detailed captions for all the figures and tables that clearly outline the main message for each of them.
* The authors should also consider other ways of adding exploration in MARL, such as enforcing diversity with different types of policies in a joint reply buffer [1].

Additionally, it would be helpful to have more details on the following:
* How generizeable do you believe your results to be? What would have to be tuned for new experiments?
* What is the F0 version in Figure 5 and Table 3?

[1] Majumdar, S., Khadka, S., Miret, S., McAleer, S. and Tumer, K., 2020, November. Evolutionary reinforcement learning for sample-efficient multiagent coordination. In International Conference on Machine Learning (pp. 6651-6660). PMLR.

**Strengths And Weaknesses:**

Strengths:
* The paper proposes a new method for MARL problem settings with tunable exploration based on the dual policy architecture.
* The analysis presented in the paper provides useful insights for how tunable exploration can be applied for different scenarios.
* The analysis of different intrinsic rewards can be useful to contribute to developing new exploration methods.

Weaknessess:
* The paper should be rewritten to fit into 12 pages with the most relevant sections and results.
* The analysis could be improved with more targeted experiments probing different questions. The intrinsic reward exploration is a good start to that, but this can be extended with more questions on the same environments or including new environments and settings. The extended analysis could further improve the paper.
* The baselines and environments studied could be better motivated, as many different MARL methods and tasks exist.

---

> ### Author Response · Authors · 2025-03-20
> **Response to Reviewer**
>
> We thank you for your review and comments. We try to address your concerns in the following points:
>
> RC1: Rewrite to fit into 12 pages
>
> Shortening the paper to fit into 12 pages is possible. However, this would require a lot of the content to be moved to the appendix.
> - part of the introduction
> - related work
> - background
> - algorithms with pseudo-code in the methods section
> - experiments
> - possibly the experiment on synchronising the exploration
>
> Given that we have added more content to further clarify our work to address other comments, we believe that shortening the paper would harm the clarity. However, we will follow the recommendation of the reviewers and action editor.
>
> RC2:  Better motivation for environments, baselines and analysis questions
>
> We have made several changes in order to improve the overall clarity of the paper.
> - We added a paragraph to the introduction where we emphasize which specific challenges are tested in the environments that we use.
> - More detail about the environment dynamics
> - Specified the challenges that are present in the LLE more clearly
> - Changed the conclusion of the results of the comparison between LRR and JRND. We now indicate that LRR shows promise in environments that contain state-bottlenecks (LLE in our experiments).
>
> RC3: More detailed captions for figures and tables
>
> We have added more information to the captions of the figures and tables, giving more context.
>
> RC4: Include other ways of exploration in the related work (e.g. Majumdar, S., Khadka, S., Miret, S., McAleer, S. and Tumer, K., 2020, November. Evolutionary reinforcement learning for sample-efficient multiagent coordination. In International Conference on Machine Learning (pp. 6651-6660). PMLR.)
>
> We have included the suggested reference in our related work as an alternative approach.
>
>
> Q1: How generizable do you believe your results to be? What would have to be tuned for new experiments?
>
> We believe that our results are generalizable to other environments. Our tests are performed on two environments that are quite different in which challenges are present, reward design, etc.  The hyperparameters used in $\epsilon$S-DP are more intuitive than the weight in weighted rewards. The most important hyperparameters are the $\epsilon$-schedules for both $\epsilon_{micro}$ and $\epsilon_{macro}$. In addition, our experiments show that the policy sampling period is an important hyperparameter. All these hyperparameters have a clear meaning and are therefore more intuitive to tune.
>
> Q2: What is the F0 version in Figure 5 and Table 3?
>
> The FO experiment in Figure 5 and Table 3 is an experiment where we modify the observation of the agents. We add the position along the axis controlled by the other agent as well as the current timestep. This makes the environment fully observable. The result of this experiment shows that the difficulty of this environment is caused by the partial observability of the environment. If the agent is able to observe the position along the axis controlled by the other agent, it can, with a recurrent neural network in its policy, extrapolate the policy of the other agent (whether it is moving towards to optimal or suboptimal reward peak) and modify its own behaviour accordingly.
> We have added this information to the caption of the figure to make this more clear.

---

### Review · Reviewer_bxJN · 2025-03-03

**Summary Of Contributions:**

This paper proposes a dual-policy architecture for multi-agent exploration, wherein exploitation and exploration are managed by separate policies. The authors introduce WQ-DP, which calculates a linear combination of the Q-values for intrinsic and extrinsic policies. Additionally, they propose $\epsilon$S-DP, which samples between exploitation and exploration policies. Experimental results demonstrate the superiority of the proposed methods.

**Audience:**

Yes

**Broader Impact Concerns:**

None.

**Claims And Evidence:**

No

**Requested Changes:**

1.	Detailed Explanations: I recommend that the authors provide detailed explanations for all mentioned concepts, especially on how the proposed method addresses the limitations discussed.

2.	Sensitivity Analysis: A sensitivity analysis of the hyperparameters is needed to demonstrate how variations affect the results.

3.	[Optional] Additional Experiments: I recommend performing experiments in the exploration scenarios of the SMAC benchmark to further demonstrate the effectiveness of the proposed method.

**Strengths And Weaknesses:**

Strengths

1.	Clear Presentation: The paper is well-organized, and the research problem of exploration using separate policies is clearly articulated.

2.	Effective Methodology: Experimental results on two benchmarks convincingly demonstrate the effectiveness of the proposed method.

Weaknesses

1.	Assumption on Intrinsic Reward: The intrinsic reward in Equation 10 plays a crucial role in exploration; however, it relies on the assumption that "reaching areas far away from the initial state" promotes exploration. The paper lacks an explanation for this assumption. It is necessary to demonstrate under what conditions this assumption holds true.

2.	Scalability in MARL: The intrinsic reward in Equation 10 is computed within the joint state space. In MARL, the joint state space becomes large, especially as the number of agents increases. In such cases, learning the Laplacian Representation is challenging.

3.	Addressing RL Issues: The authors mention issues in RL, such as noisy-TV, interdependence, and perfect coordination, in Section 1. However, it is unclear how the proposed method overcomes these limitations.

4.	Control Mechanism of $\epsilon$S-DP: In Section 4.1.2, the authors claim that $\epsilon$S-DP can control "how often" to switch between the two policies, but it is not clear how this control is achieved.

5.	Lack of Sensitivity Analysis: The paper relies on several hyperparameters but lacks a sensitivity analysis to show how results change with different hyperparameter settings.

6.	Formal Definition Needed for $\epsilon$S-DP: $\epsilon$S-DP needs to be formally defined with a mathematical equation. The term “similar to ϵϵ-greedy” may lead to confusion.

7.	[Minor] Explanation of Concepts in Figures: Some concepts in the figures need explanation, such as "Policy Mixing" in Figure 2.

8.	[Minor] Clarification of Notations: Some notations need to be explained, such as $s_{goal}$ Equation 8.

---

> ### Author Response · Authors · 2025-03-20
> **Response to Reviewer**
>
> Thank you for your valuable feedback on our work. We addressed your comments in the following way:
>
> RC1: Detailed explanation for all concepts
>
> We have added several elements to improve the clarity of the paper:
> - We added a sentence acknowledging the possible scalability concerns when using an intrinsic reward calculated using the joint observation (W2).
> 		"However, in environments with a large number of agents there may be scalability issues when using the joint observation. In this case, an alternative approach may be required."
> - Algorithms with pseudo-code for each of the architectures that we compare. This makes it clear which aspects differ between each of the approaches and gives a more formal definition of our approaches (W4&6).
> - More detail about the environments.
> - A paragraph in the introduction where we emphasize which specific challenges are tested in the environments that we use.
> - Highlighted in our conclusion that LRR shows promise in environments that contain bottlenecks in the state-space (W1).
> - We clarify the concepts mentioned in W7 and W8.
>
> The dual-policy architectures are not specifically designed to tackle one specific challenge. But, by testing the approach on environments that test for these challenges we are able to show that the benefits of our approach are widely applicable. We also added a paragraph in the conclusion highlighting this. (W3)
>
> RC2: A sensitivity analysis of hyperparameters
>
> In our experiments, we show the impact of changing two parameters within the $\epsilon$S-DP approach. These experiments clearly show the impact of varying the value of the policy sampling period and adapting the synchronisation of the exploration between the agents. One aspect that could be investigated further is the schedule of $\epsilon_{macro}$ (and $\epsilon_{micro}$). However, given that the schedule can be adapted in so many ways, this would warrant a dedicated study on the impact of this parameter which falls outside the scope of this work.
>
> RC3: Additional experiments
>
> We chose the current set of environments based on their explicit focus on exploration. The environments are designed to be challenging exploration tasks.
> As explained by Molinghen et al. [1], this is not the case for the most commonly used benchmarks such as SMAC. In SMAC, individual agents are often able to explore the entire environment without cooperation with other agents and can sometimes even achieve the goal on their own. Therefore, we believe that the two environments we used are the most relevant for our work.
>
> [1] Molinghen, Y., Avalos, R., Van Achter, M., Nowé, A., & Lenaerts, T. (2023, November). Laser Learning Environment: A new environment for coordination-critical multi-agent tasks. In _Benelux Conference on Artificial Intelligence_ (pp. 135-154). Cham: Springer Nature Switzerland.

---

> > ### Comment · Reviewer_bxJN · 2025-04-17
> >
> > Thank you for your response. However, the current response does not fully address all of my concerns. Below, I outline the key issues that still require attention:
> >
> > 1. The authors mention potential weaknesses (W1, W2) in the revised manuscript but do not propose any effective approaches to address these issues.
> >
> > 2. Since the authors discuss different types of challenges in the main body of the paper, corresponding environments and experiments need to be carefully designed to address these challenges (W3).
> >
> > 3. A sensitivity analysis is still lacking (W5).
> >
> > I hope the authors can address these concerns in their next revision.

---

> > > ### Author Response · Authors · 2025-04-25
> > >
> > > Thank you for the additional feedback.  We have marked the changes addressing your comments in red in the current revision.
> > >
> > > 1. In the current revision we have included some additional information regarding these potential downsides:
> > >
> > > 	W1: This assumption is inherent to the proposed LLR. We acknowledge that this assumption does not hold in every environment (e.g. MPE where the agents are not bound within a certain area and can move infinitely far away). However, in certain environments with specific dynamics such as bottlenecks in the state-space, LLR can help to overcome these challenging dynamics. Therefore, it is important to evaluate the environment in advance in order to determine whether using the LRR will be effective. We have modified our text in order to make this more clear.
> > >
> > > 	 W2: The scalability issues that arise in this context can be mitigated by calculating an individual intrinsic reward through an individual Laplacian representation. This would reduce the ability of the agents to coordinate their exploration. Therefore, there will always be a trade-off between the scalability of the reward calculation and the ability of the agents to jointly explore the state-action space.
> > >
> > >  2.  In our experiments, we use environments that are specifically designed to test the performance of the agents when facing the challenges we have explained. The rel_overgen environment specifically focuses on relative overgeneralisation and high interdependence. The LLE challenges the agents with sparse rewards, bottlenecks in the state-space, zero-incentive dynamics, high interdependence and requires perfect coordination. Therefore, our experiments are able to show how each of the tested configurations is able to deal with these challenges.
> > >
> > > 3. Our work currently investigates two of the introduced hyperparameters within $\epsilon$S-DP. Apart from those, the only other hyperparameter that is unique to $\epsilon$S-DP is the $\epsilon_{macro}$ value and its schedule. We believe that in order to be able to provide comprehensive insight into the influence of this parameter, a separate, in-depth work is required since many different schedules are possible. Therefore, we have included this in the current revision in our future work section. Apart from this we do not see for which hyperparameter an additional analysis would be useful.
> > >
> > > We hope this addresses all of your remaining concerns.

---

### Review · Reviewer_1yJ7 · 2025-03-15

**Summary Of Contributions:**

This paper proposes a dual-policy architecture for multi-agent reinforcement learning (MARL) that separates exploration and exploitation behaviors. The authors introduce two specific implementations: Weighted-Q Dual-Policy (WQ-DP) and $\epsilon$-Sampled Dual-Policy (εS-DP). Unlike traditional approaches that combine exploration and exploitation into a single policy via weighted rewards, these dual-policy methods allow each policy to focus on maximizing either the environment reward or the intrinsic exploration reward.

The authors evaluate their approach on two environments: a relative overgeneralization environment and the Laser Learning Environment (LLE). They demonstrate that both dual-policy approaches outperform traditional baselines, with $\epsilon$S-DP showing the best overall performance. The paper also introduces a new intrinsic reward based on Laplacian representations (LRR) designed to encourage agents to move away from initial states and overcome bottlenecks.

Additionally, the authors analyze several aspects of the $\epsilon$S-DP architecture, including policy sampling period and synchronization of exploration across agents, providing insights into what factors contribute most to performance improvements.

**Audience:**

Yes

**Claims And Evidence:**

Yes

**Requested Changes:**

- Include comparisons with more recent and relevant MARL exploration baselines, particularly MAVEN and "Celebrating Diversity in Shared MARL". These methods specifically address exploration in MARL and represent important benchmarks.
- Evaluate the approach in at least one more complex MARL benchmark environment such as SMAC or Multi-Agent MuJoCo. This is essential to demonstrate that the benefits extend beyond gridworld scenarios.
- Design and include experiments that specifically test the challenges mentioned in the introduction (particularly the Noisy-TV problem and sparse rewards). Currently, there's a disconnect between the motivation and the empirical validation.

**Strengths And Weaknesses:**

# Strengths:
- The paper addresses an important problem in MARL: separating exploration from exploitation to handle the multi-objective nature of learning.
- The proposed dual-policy architecture is conceptually clear and well-motivated from a theoretical perspective.
- The empirical analysis of policy sampling period is insightful and demonstrates the value of allowing policies to operate for extended periods.
- The introduction of Laplacian Representation Reward shows promise for overcoming state-space bottlenecks.
- The paper is well-written and easy to follow.

# Weaknesses:
- The baseline comparisons are inadequate. The paper fails to compare against significant state-of-the-art MARL exploration methods like MAVEN, "Celebrating Diversity in Shared Multi-Agent Reinforcement Learning", etc. This makes it difficult to assess the true contribution relative to recent advances.
- Despite the introduction mentioning numerous exploration challenges (sparse rewards, Noisy-TV, bottlenecks, etc.), the experimental evaluation doesn't directly demonstrate how the proposed methods address most of these challenges. This creates a disconnect between the claimed benefits and the empirical validation.
- The experimental environments are limited to relatively simple gridworld-like environments. The absence of evaluations on more complex benchmarks like SMAC (StarCraft Multi-Agent Challenge) significantly limits the generality of the claims.
- The paper doesn't clearly demonstrate how the proposed methods scale to environments with larger numbers of agents or more challenging SMAC domains.
- There's insufficient analysis of computational overhead introduced by maintaining dual policies compared to traditional approaches.

---

> ### Author Response · Authors · 2025-03-20
> **Response to Reviewer**
>
> Thank you for your comments and feedback. In the following points we try to address your concerns:
>
> RC1: Include comparisons with other exploration baselines (MAVEN, etc.)
>
> The focus of our work is to showcase the benefits of splitting the exploitation and exploration policies. Our goal is not necessarily to outperform all other exploration approaches. Our focus is to show that our approaches outperform the alternative when using an intrinsic reward, WR-SP. Therefore, we choose baselines that fit within this goal, $\epsilon$G-DP and WR-SP.  Other baselines such as MAVEN stray too far from the approaches where our architecture is useful. MAVEN uses a shared latent variable that allows the agents to coordinate the behaviour they will try during that episode. This is a fundamentally different approach than what we focus on, learning exploration through an intrinsic reward.
>
> RC2: Evaluate on complex MARL benchmark (SMAC, Multi-Agent MuJoCo)
>
> We chose the current set of environments based on their explicit focus on exploration. The environments are designed to be challenging exploration tasks.
> As explained by Molinghen et al. [1], this is not the case for the most commonly used benchmarks such as SMAC. In SMAC, individual agents are often able to explore the entire environment without cooperation with other agents and can sometimes even achieve the goal on their own. Therefore, we believe that the two environments we used are the most relevant for our work.
>
> RC3: Include experiments specifically testing the mentioned challenges
>
> In our experiments, the used environments test on a large part of the mentioned challenges. The LLE environment has high interdependence, required perfect coordination, has zero-incentive dynamics and bottlenecks in the state-space and includes sparse rewards. The rel_overgen environment is specifically designed to test agents on relative overgeneralisation. The environment also has high interdependence.
> Apart from the noisy-TV problem, all mentioned challenges are present in the environments we used.
>
> [1] Molinghen, Y., Avalos, R., Van Achter, M., Nowé, A., & Lenaerts, T. (2023, November). Laser Learning Environment: A new environment for coordination-critical multi-agent tasks. In _Benelux Conference on Artificial Intelligence_ (pp. 135-154). Cham: Springer Nature Switzerland.

---

### Author Response · Authors · 2025-03-20
**General response to the reviewers**

We sincerely thank all reviewers for their valuable and insightful feedback. We made a number of changes in the current revision in order to address the provided comments and improve the overall clarity regarding the feedback of the reviewers.

- We included an algorithm with pseudo-code for each of the tested architectures. This improves the clarity of what the differences are between each of the approaches.
- We added a paragraph in the introduction to clarify which challenges are present in the environments that we use.
		"In our experiments, we use environments that are designed to test the ability of our approaches to deal with the multi-agent exploration challenges that we explained as well as several of the single-agent challenges. The laser learning environment challenges the agents with sparse rewards, bottlenecks in the state-space, zero-incentive dynamics, high interdependence and requires perfect coordination while the relative overgeneralisation environment focuses on relative overgeneralisation and high interdependence."
- We added a paragraph in the conclusion giving more context into how our approach influences the challenges presented in the introduction
		"We evaluate on two challenging environments that exhibit many exploration challenges. Our results show that through the use of our architectures, the exploration policies are able to perform better and deal with these challenges more effectively. By separating exploration and exploitation, both policies are able to focus on a single task and this allows them to learn more successful behaviours.
- We added a figure to the appendix giving more details about the observation provided by the LLE.
- We added a figure that shows the difference between L2 distance and a distance that is based on the number of steps it takes to reach a location. This shows that a distance metric that takes into account the environment dynamics (as in LRR) provides a much more insightful intrinsic reward than L2 distance.
- We improved the clarity of the conclusion regarding the LRR results.
		"...the results show the potential of using Laplacian representations to calculate an intrinsic reward in environments that contain bottlenecks in the state-space such as the LLE."
- We made a number of small changes throughout our work to improve clarity.

---

### Decision · Action_Editor_FP6i · 2025-04-28

**Recommendation:** Reject

**Comment:**

All reviewers agree on the merit of this work, but all reviewers agree that it is not ready for publication in its current form. I would recommend the authors address all the feedback provided by the reviewers and re-submit a new version of this work.

**Audience:**

There is certainly an audience for this work. However, as pointed out by reviewer 7tWo, the length of the paper (currently at 24 pages) may limit the readership and impact of this work. I would recommend the authors consider shortening the work. There are plenty of publications of a similar scope that fit in under 12 pages, so it is certainly possible to do so without sacrificing clarity.

**Claims And Evidence:**

The experiments provided do a good job of highlighting some of the qualities of the proposed method and, while promising, are not yet mature enough for publication.

First of all, they fall a bit short of properly situating in the relevant literature. I would suggest the authors consider benchmarking against the methods and benchmarks suggested by reviewers 1yJ7 and 7tWo, as it helps properly position the proposed method against existing methods.

Secondly, as pointed out by reviewer bxJN, it would be good to empirically demonstrate how to overcome some of the weaknesses discussed as opposed to simply discussing them.

**Resubmission Of Major Revision:**

The authors may consider submitting a major revision at a later time.